# Modified *t*-butyl in tetradentate platinum (II) complexes enables exceptional lifetime for blue-phosphorescent organic light-emitting diodes

Young Hun Jung[1,3], Gyeong Seok Lee[2,3], Subramanian Muruganantham [1], Hye Rin Kim[1], Jun Hyeog Oh[1], Jung Ho Ham[1], Sagar B. Yadav[1], Ji Hyun Lee[2], Mi Young Chae[1], Yun-Hi Kim [2] ✉ & Jang Hyuk Kwon [1] ✉

In blue phosphorescent dopants, the tetradentate platinum(II) complex is a promising material showing high efficiency and stability in devices. However, metal-metal-to-ligand charge transfer (MMLCT) formation leads to low photoluminescence quantum yields (PLQYs), wide spectra, and intermolecular interaction. To suppress MMLCT, PtON-tb-TTB and PtON-tb-DTB are designed using theoretical simulation by modifying t-butyl in PtON-TBBI. Both materials effectively suppress MMLCT and exhibit high PLQYs of 99% and 78% in 5 wt% doped film, respectively. The PtON-tb-TTB and PtON-tb-DTB devices have maximum external quantum efficiencies of 26.3% and 20.9%, respectively. Additionally, the PtON-tb-DTB device has an extended lifetime of 169.3 h with an initial luminescence of 1200 nit, which is 8.5 times greater than the PtON-TBBI device. Extended lifetime because of suppressed MMLCT and smaller displacement between the lowest triplet and triplet metal-centered states compared to other dopants. The study provides an effective approach to designing platinum(II) complexes for long device lifetimes.

Organic light-emitting diodes (OLEDs) have been utilized for several applications in various displays over the past 25 years[1]. However, the current technology for OLED displays aims to develop devices with higher efficiency, color purity, and longer operational stability, but it is still elusive. Specifically, red, and green phosphorescent materials were successfully commercialized in the market due to their highly efficient electroluminescence and longer device lifetime. In contrast, blue OLEDs employed low-efficiency fluorescent dopants due to poor device lifetimes of phosphorescent and thermally activated delayed fluorescence (TADF) OLEDs[2–5]. Much intensive research is still being done to find ways to extend the device durability of blue phosphorescent or TADF OLEDs.

A great deal of conventional blue TADF emitters have been explored because of mild synthetic difficulty, low-efficiency roll-off, and acceptable device lifetime owing to triplet exciton utilization. Kwon et al. recently revealed that the stable blue TADF emitter DBA-DI had a long lifetime ($LT_{50}$) of 540 h at an initial brightness of 1000 nits and a high EQE of 26.4%[6]. It was found that electron traps, which cause material degradation in TADF materials should be reduced through host engineering. Adachi et al. reported TADF OLED with a long lifetime $LT_{95}$ of 29 h at an initial luminescence of 1000 nits and a high EQE of 22% by employing HDT-1 dopant[7], which is intended to inhibit aggregation effects through the *m*-terphenyl units. Blue TADF OLEDs have been observed to exhibit poorer device stability in comparison to

[1]Organic Optoelectronic Device Lab (OODL), Department of Information Display, Kyung Hee University, Seoul, Republic of Korea. [2]Department of Chemistry and RIMA, Gyeongsang National University, Jinju, Republic of Korea. [3]These authors contributed equally: Young Hun Jung, Gyeong Seok Lee. ✉e-mail: ykim@gnu.ac.kr; jhkwon@khu.ac.kr

phosphorescent OLEDs. This is because the TADF material's strong intramolecular charge transfer characteristics lead to red-shifted emission behavior, which in turn results in higher bandgap and triplet energy characteristics. These higher energy values can cause some reduction in the device stability of blue TADF OLEDs[6,8]. As a result, phosphorescence and TADF materials have emerged as potential candidates for deep blue OLEDs because of their longer device lifetime and higher color purity. Later, Choi et al. reported a long-lifetime blue phosphorescent OLED employing the CN-Ir(III) dopant[8]. This device exhibited LT$_{95}$ of 232 h at an initial luminescence of 500 nits with a maximum EQE of 25.1% by using an optimized exciplex host combination system, which could have good charge balance and efficient energy transfer characteristics. However, Ir(III) complexes had low color purity caused by broad spectrum and showed second order vibrational peak. Therefore, recently square planar Pt(II) complexes have been researched which are attributed to high device stability and narrow full width at half maximum (FWHM).

The square planar tetradentate Pt(II) complexes comprising the primary ligands of the carbazolyl-pyridine N-heterocyclic carbene (NHC) as a highly rigid structure induces higher PLQY and improves the anticipated electroluminescence performance. However, these Pt(II) complexes were significantly affected by MMLCT (Metal-Metal to Ligand Charge Transfer) formation which originated from orbital overlapping of vacant $d_{z^2}$ orbitals[9–25]. This behavior can lead to unwanted triplet exciton diffusion through Dexter energy transfer (DET)[26–29]. Multi-step exciton diffusion between Pt(II) complexes arises from the pathways of triplet-triplet annihilation (TTA) and triplet-polaron annihilation (TPA), which induce material degradation in the device[29,30]. In the previous reports, Li et al. examined the spectrum broadening of the substitution position in PtON1 and PtON7, which are composed of ancillary ligands that are either phenyl pyrazole, methylimidazole, or tetradentate cyclometalated ligands. PtON1 and PtON7, without any additional substitution on the ligand's motif, can result in drastically broad PL spectra. Further, this issue has been addressed by introducing alkyl and dimethyl amine donating substitutions on the *para*-position of the pyridine unit in the primary NHC ligand, and it shows a narrower spectrum, which effectively suppresses the state mixing between $^1$MLCT (metal-to-ligand charge transfer), $^3$MLCT, and $^3$LC (ligand center) states[31]. Later, Kim et al. introduced the adamantyl group on the *para*-position of pyridine to decrease MMLCT formation, which leads to a narrow spectrum in solution and film states[32]. However, alkyl substitutions such as methyl and adamantyl groups have an unsuitable application as substituents owing to their insufficient bulkiness or high molecular weight compared to the *t*-butyl group, respectively. On the other hand, Li et al. have reported PtON7-tBu, which shows a narrow spectrum with bulky substitution of the *t*-butyl group incorporated on the primary ligand pyridine unit *para*-position. In addition, the *t*-butyl group substituted on the *meta*-position of the ether linkage phenyl group on PtON7-dtb which is attributed to suppressed MMLCT formation. It can improve color purity in the film state, although it shows enhanced shoulder peak in the solution state.

The abovementioned tendency was used by the PtON ligand to develop PtON7-tBu blue emitters which showed high color purity and forcibly inhibited molecular concentration quenching. Despite the imidazolium carbene ligands-based Pt(II) complexes, they do not perform well in the device stability because of the MMLCT formation and activated non-radiative decay process. Recently, Kim et al. discovered benzimidazolium carbene moiety of the tetradentate Pt(II) complexes PtON-TBBI to increase PLQY due to elongated π-conjugation and promote the rigidity of the excited state[33]. Remarkably, the addition of bulky substituents of 3,5-di-*tert*-butylphenyl attached to the benzimidazolium carbene moiety restricts the molecular stacking and induces more steric hindrance on the benzimidazolium carbene unit that facilitates to increase the,

$E_{a,T1 \rightarrow 3MC}$ with high PLQY, which is highly associated with a good triplet energy confinement and narrow FWHM of 24 nm. Subsequently, the PtON-TBBI dopant was studied with LT$_{95}$ of 150 h, initial luminescence of 1000 nits, and maximum EQE of 25.4%. PtON-TBBI has a shorter exciton lifetime than PtON7-tBu, attributed to the increased MLCT characteristics. Hence, the tetradentate Pt(II) complex has garnered a lot of attention for its outstanding performance and has become one of the most stable blue phosphorescent dopants. As a result, modifying the substitution position and increasing the π-conjugated ligand allowed for fine-tuning the photophysical characteristics of the tetradentate NHC-based Pt(II) complex. Although photo-physical property changes of the materials are well documented using *t*-butyl groups on the primary PtON ligand, the substitution of *t*-butyl group influences to conduct the device's lifetime is not explored.

In this study, we introduce two new Pt(II) complexes, PtON-tb-DTB and PtON-tb-TTB, designed via theoretical simulation methods. The design involved adding or removing bulky *t*-butyl groups in different substitution positions on the primary PtON ligand. This allowed us to investigate the structural relationship between the substituent position effects and the suppression of MMLCT formation. These Pt(II) complexes of PtON-tb-DTB and PtON-tb-TTB exhibited high PLQYs of 78% and 99% in the 5 wt% doped PMMA film, respectively. On the other hand, the fabricated OLEDs using PtON-tb-TTB achieved a maximum EQE value of 26.7%, which was higher than that of 25.9% PtON-TBBI. Notably, PtON-tb-DTB demonstrated a significantly extended lifetime of 169.3 h at an initial luminescence of 1200 nits in the PhOLED device, compared to the inadequate 20 h lifetime of PtON-TBBI. This improvement can be attributed to the decreased TTA process and better hot exciton stability of blue phosphorescent OLEDs. Factors such as MMLCT formation, lower activation energy from 3MC to $T_1$ ($E_{a,3MC \rightarrow T1}$), and small geometrical changes between 3MC and $T_1$ states were considered and analyzing these findings.

## Results

### Material design concept and theoretical calculation for platinum(II) complexes

The primary focus of this study involves modifying Pt(II) complexes to develop stable blue phosphorescent emitters suitable for OLED displays. The changes of the ancillary ligands were considered to minimize molecular interaction, so-called MMLCT, for decreasing excited state interaction. To tackle this issue we decided to design two new Pt(II) complexes, namely PtON-tb-DTB and PtON-tb-TTB, by modulating their bulky *t*-butyl groups substituted in distinct positions. These Pt(II) complexes were designed via molecular dynamics (MD) and quantum chemistry (QC) simulation methods, as illustrated in Fig. 1. The stable new Pt(II) complex has been designed by two main strategies: (i) by introducing a bulky substitution of the *t*-butyl group on the *meta* position of the ether linkage phenyl ring, resulting in a negligible red-shifted PL and enhancing the vibrational peak; and (ii) with PtON-tb-DTB, one *t*-butyl group is removed from the benzimidazolium carbene substituted phenyl ring, which reduces steric hindrance and makes the moiety more flexible and rotatable, which can effectively hinder the formation of MMLCT and reduce the $E_{a,3MC \rightarrow T1}$. This indicates that because of the lower steric hindrance, PtON-tb-DTB has a more distorted conformation than PtON-TBBI through the freely rotating motion of bulky substitution. The PtON-tb-DTB possesses a larger dihedral angle at the $T_1$ state, which is expected to alleviate the formation of the MMLCT by reducing the intermolecular interaction between the vacant $d_{z^2}$ orbitals of Pt(II) (central metal atom). Thus, the formation of MMLCT could be more effectively suppressed by PtON-tb-DTB than that of the PtON-TBBI dopant. Thus, we believe that this prominent design strategy can help to enhance the device performance compared to a previously reported complex PtON-TBBI,

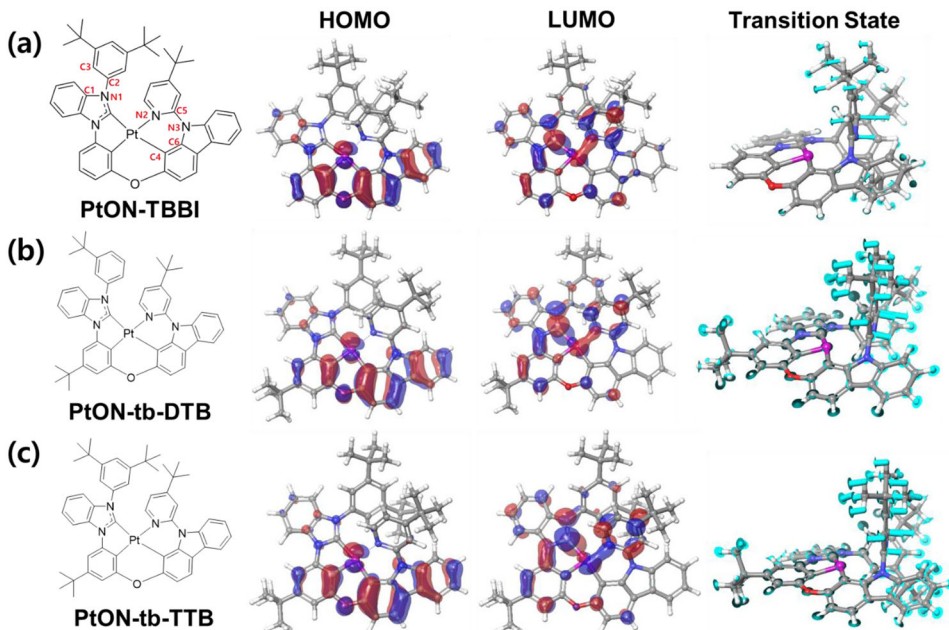

**Fig. 1 | Calculated HOMO-LUMO and transition state between T1 and 3MC state. a** PtON-TBBI, **b** PtON-tb-DTB, and **c** PtON-tb-TTB via Quantum Chemical (QC) Simulation.

respectively. Further to clarify the changes in HOMO-LUMO energy levels, and molecular conformation changes, QC simulation is conducted for PtON-TBBI, PtON-tb-DTB, and PtON-tb-TTB. As illustrated in Fig. 1, the highest occupied molecular orbital (HOMO), lowest unoccupied molecular orbital (LUMO) and transition state between $T_1$ and 3MC state. The Calculated HOMO/ LUMO energy levels of PtON-TBBI, PtON-tb-DTB, and PtON-tb-TTB are −5.57/ −2.40 eV, −5.52/ −2.44 eV, −5.54/ −2.43 eV, respectively. It is worth noting that PtON-tb-TTB with a *t*-butyl substituent on the *meta*-position of the ether linkage phenyl ring attributed shallower HOMO and deeper LUMO than PtON-TBBI because of the hyperconjugation effect[34]. In addition, PtON-tb-DTB with one *t*-butyl group removed on the benzimidazolium carbene substituted phenyl ring also has shallower HOMO and deeper LUMO than PtON-TBBI. Moreover, reduced steric hindrance and decreased dihedral angles (∠C1-N1-C2-C3) of PtON-TBBI, PtON-tb-DTB, and PtON-tb-TTB are 56.5°, 54.7°, and 56.4° respectively. PtON-tb-DTB's reduced ∠C1-N1-C2-C3 enhanced conjugation in contrast to the other complexes such as PtON-TBBI and PtON-tb-TTB, respectively. It was clarified through the bond length of N1-C2[35]. The N1-C2 values of PtON-TBBI, PtON-tb-DTB, and PtON-tb-TTB are 1.421, 1.417, and 1.420 Å, respectively. Although energy band gaps of the new materials are smaller than that of PtON-TBBI owing to increased conjugation, those values can be negligible as we expected.

Afterwards, MD simulation was used to determine the intermolecular distance in the film state. For PtON-TBBI, PtON-tb-DTB, and PtON-tb-TTB, the corresponding number densities are 0.901, 0.904, and 0.811/ $nm^3$, respectively. As a result, the additional *t*-butyl group on the meta position of the ether linkage phenyl ring significantly increases the intermolecular distance as compared to PtON-TBBI. On the other hand, there was minimal variation in the numerical density between PtON-tb-DTB and PtON-TBBI. The dihedral angles (∠C4-Pt-N2-C5) of PtON-TBBI, PtON-tb-DTB, and PtON-tb-TTB are 15.5°, 20.3°, and 16.0°, respectively. This indicates that because of lower steric hindrance, PtON-tb-DTB has more distorted conformation than PtON-TBBI through freely rotation motion of substitution. The PtON-tb-DTB possesses a larger dihedral angle at the $T_1$ state, which is expected to alleviate the formation of the MMLCT by reducing the intermolecular interaction between the vacant $d_{z^2}$ orbital of Pt(II) (central metal atom). Thus, compared to PtON-TBBI dopant, the MMLCT formation

could be suppressed more successfully in PtON-tb-DTB. Furthermore, a QC simulation was performed on PtON-tb-MTB to elucidate the changes in $T_1$ geometry and transition state following the replacement of the *t*-butyl group with a phenyl ring on the benzimidazolium carbene ligand. The calculated ∠C1-N1-C2-C3 and ∠C4-Pt-N2-C5 values of PtON-tb-MTB at $T_1$ states are 42.5° and 23.5°, respectively. It means that the free rotation motion due to the removal of the *t*-butyl group can affect the geometrical change in the $T_1$ state. The angle between PtON-tb-MTB and ∠C6-N3-C5-N2 in the transition state is −109.6°, which is lower than the value of −90.7° for PtON-tb-DTB. This shows that the geometry of the transition state can be adjusted by adding the *t*-butyl group to the benzimidazolium carbene substituted phenyl ring. This is because of the steric hindrance between the *t*-butyl group on the phenyl ring and the pyridine moiety. As a result, the PtON-tb-MTB may not be a good desired molecule, although it has a higher ∠C4-Pt-N2-C5 due to significantly reduced steric-hindrance, which can cause severe vibrational relaxation and the simulation results are presented in Supplementary Fig. 29[16].

In addition, $E_{a,T1\rightarrow 3MC}$ is considered one of the most essential parameters for determining PLQYs. Once it has a high barrier of this activation energy, it will confine easily triplet excitons at the excited state. It should be highly encouraged to suppress the transition from $T_1$ to the 3MC state because the 3MC state acts as the non-radiative center. The calculated $E_{a,T1\rightarrow 3MC}$ values for PtON-TBBI, PtON-tb-DTB, and PtON-tb-TTB are 0.821, 0.827, and 0.846 eV, respectively. These three materials showed high $E_{a,T1\rightarrow 3MC}$ values, but comparatively, PtON-tb-TTB had the highest $E_{a,T1\rightarrow 3MC}$ value which induced the highest PLQY. Even though PtON-tb-DTB and PtON-TBBI have similar $E_{a,T1\rightarrow 3MC}$ values, PtON-tb-DTB with freely rotatable moiety can have lower PLQY due to vibrational relaxation. Additionally, the $E_{a,3MC\rightarrow T1}$ values of the materials were calculated by QC simulation method, appeared activation energy to transfer from 3MC to $T_1$ state. The calculated $E_{a,3MC\rightarrow T1}$ values for PtON-TBBI, PtON-tb-DTB, and PtON-tb-TTB are 0.509, 0.423, and 0.456 eV, respectively. Based on these findings, it appears that both novel materials may be easier to transition from 3MC to the $T_1$ state because of its lower energy barrier. Especially, PtON-tb-DTB has the smallest energy barrier, which means that rapidly transition to the $T_1$ state. QC and MD simulation results are summarized in Table 1.

## Material synthesis and characterization

In our synthesis, the material design strategy implementation of new Pt(II) complexes, PtON-tb-TTB and PtON-tb-DTB, was successfully done. The target materials were synthesized by the following methods outlined in Supplementary Fig. 1. The detailed synthetic procedure and the related data are provided in the supporting information. Briefly, 1-(3-bromo-5-(tert-butyl)phenyl)−1H-benzo[d]imidazole and (tert-butyl) pyridylcarbazole moiety were used to prepare the key intermediate **1** and **3**. The tetradentate ligands **2, 4** and **5** were prepared by Cu(OAc)₂-catalyzed formation between 2-(3-(1H-benzo[d]imidazol-1-yl)−5-(tert-butyl)phenoxy)−9-(4-(tert-butyl)pyridin-2-yl)−9H-carbazole and (3,5-di-tert-butylphenyl)(mesityl)iodonium triflate and (3-(tert-butyl)phenyl)(mesityl)iodonium triflate. Finally, tetradentate Pt(II) complexes, PtON-tb-TTB and PtON-tb-DTB, were synthesized by the cyclometallation of ligands **2, 4** and **5** with dichloro(1,5-cyclooctadiene)platinum(II) (Pt(COD)Cl₂) under reflux condition with the isolated yields of 49-52%. All the intermediates, ligands, and desired target Pt(II) complexes were purified by silica gel column chromatography method and the desired products were systematically characterized and identified by ¹H and ¹³C nuclear magnetic resonance (NMR) spectroscopy and high-resolution mass spectrometry (HRMS-QTof) analyses (the detailed synthetic scheme, procedure and characterization were explained in the Supplementary Figs. 2-19).

## Photo-physical property

UV-visible and PL spectroscopy were used to measure the absorption and photoluminescence (PL) spectra as shown in Fig. 2a. The wavelengths at the maximum intensity ($\lambda_{PL}$) of PL spectra of PtON-TBBI, PtON-tb-DTB, and PtON-tb-TTB are 452.8, 457.8, and 457.0 nm, respectively. The HOMO and LUMO energy levels are calculated using the optical bandgap and oxidation potential obtained from the onset wavelength of the absorption spectrum and cyclic voltammetry (CV) measurement. Supplementary Fig. 22 shows the oxidation potential

curves. The measured HOMO/LUMO values of PtON-TBBI, PtON-tb-DTB, and PtON-tb-TTB are −5.55/−2.70 eV, −5.50/−2.71 eV, and −5.53/−2.71 eV, respectively. Both the PL spectra and HOMO/LUMO data show a similar trend as expected by the QC simulation. Upon adding the t-butyl group at the meta-position of the ether linkage phenyl ring, LUMO of PtON-tb-TTB is lower than PtON-TBBI due to hyperconjugation. On the other hand, removing one of the t-butyl groups from the 3,5-di-tert-butyl-phenyl group also increases conjugation thus reducing steric hindrance. Consequently, both new materials have a shallower HOMO and deeper LUMO than PtON-TBBI, especially PtON-tb-DTB has the shallowest HOMO. The difference in shoulder peaks which are affected by vibrational bands is observed in the PL spectrum. To compare the spectra according to the substitution group, the spectra of PtON-tb-DTB and PtON-tb-TTB are shifted to that of PtON-TBBI. Figure 2b clearly shows the spectral shift of PtON-TBBI, PtON-tb-TTB and PtON-tb-DTB, which exhibited elevated second and third vibronic peaks with the addition of t-butyl on the ether phenyl ring. As shown in Fig. 2c, time-resolved photoluminescence (TRPL) spectra were used to measure exciton lifetime ($\tau_d$) by doping materials with a poly(methyl methacrylate) (PMMA) matrix. The exciton lifetime ($\tau_d$) of 5 wt% doped PtON-TBBI, PtON-tb-DTB, and PtON-tb-TTB are 2.64, 3.25, and 2.97 μs, respectively. Clearly, the new materials exhibit longer lifetimes than that of PtON-TBBI. The radiative ($k_r^T$) and non-radiative triplet rate ($k_{nr}^T$) were calculated using the PLQY through a simple relation[33]:

$$PLQY = \frac{k_r^T}{k_r^T + k_{nr}^T} = \frac{k_r^T}{k_d} \tag{1}$$

In addition, the PLQYs in thin film of PtON-TBBI, PtON-tb-DTB, and PtON-tb-TTB were estimated to be 0.95, 0.78, and 0.99, respectively. Although the activation energy of PtON-TBBI and PtON-tb-DTB are similar, there is a great difference of 0.17 in PLQY. Decreased steric hindrance by removing one t-butyl group increases the vibrational

## Table 1 | Summary of calculated outputs from QC/MD simulation

|  | Calculated HOMO (eV)[a] | Calculated LUMO (eV)[a] | ∠C1-N1-C2-C3 (°) | ∠C4-Pt-N2-C5 (°) | $E_{a,T1→3MC}$ (eV)[b] | $E_{a,3MC→T1}$ (eV) | Number density (/nm³)[c] |
|---|---|---|---|---|---|---|---|
| PtON-TBBI | −5.57 | −2.40 | 56.5 | 15.5 | 0.821 | 0.509 | 0.901 |
| PtON-tb-DTB | −5.52 | −2.44 | 54.7 | 20.3 | 0.827 | 0.423 | 0.904 |
| PtON-tb-TTB | −5.54 | −2.43 | 56.4 | 16.0 | 0.846 | 0.456 | 0.811 |

[a]Calculated value through QC simulation by using B3LYP-D3 functional/ LACV3P + +** basis-set.

[b]Calculated values through QC simulation by using QST method.

[c]Calculated values through MD simulation.

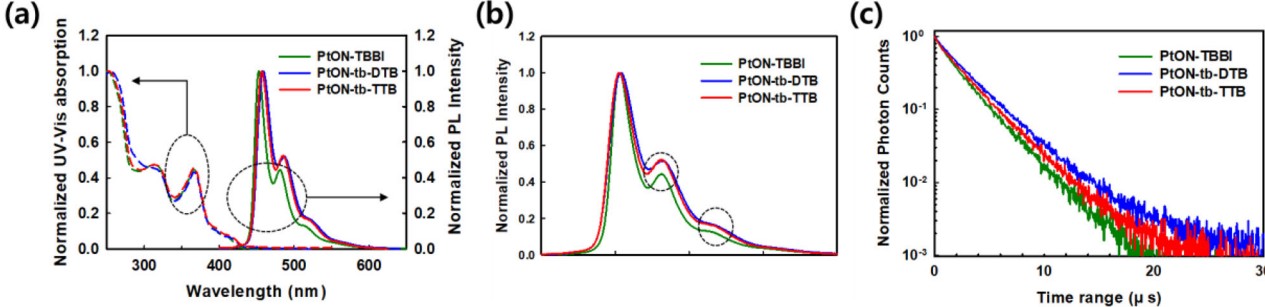

**Fig. 2 | The photo-physical measurement. a** UV−visible spectroscopy in methylene chloride (1.0 × 10⁻⁵ M) and photoluminescence spectroscopy in 5 wt% doped PMMA of PtON-TBBI, PtON-tb-DTB, and PtON-tb-TTB. **b** Shifted spectra of PtON-tb-DTB and PtON-tb-TTB to that of PtON-TBBI for comparison of the vibrational bands. **c** Measurements of time-resolved photoluminescence (TRPL) spectra by using 5 wt% doped on the PMMA matrix.

**Table 2 | Photo-physical properties obtained from experiments**

| | $E_g$ (eV)[a] | HOMO (eV)[b] | LUMO (eV)[b] | $\lambda_{PL}$ (nm)[c] | $\tau_d$ (µs)[d] | $k_r^T$ (/s)[e] | $k_{nr}^T$ (/s)[f] |
|---|---|---|---|---|---|---|---|
| PtON-TBBI | 2.85 | −5.55 | −2.70 | 452.8 | 2.64 | $3.60 \times 10^5$ | $1.89 \times 10^4$ |
| PtON-tb-DTB | 2.79 | −5.50 | −2.71 | 457.8 | 3.25 | $2.40 \times 10^5$ | $6.77 \times 10^4$ |
| PtON-tb-TTB | 2.82 | −5.53 | −2.71 | 457.0 | 2.97 | $3.33 \times 10^5$ | $3.37 \times 10^3$ |

[a]Optical bandgap calculated by using on-set wavelength on absorption spectra ($E_{g,optical} = \frac{1240}{\lambda_{on}}$).
[b]Measured HOMO and LUMO energy level. HOMO energy levels are calculated through CV measurements, and LUMO are calculated by adding optical bandgap and HOMO energy level.
[c]Maximum emission wavelength in the PL spectrum.
[d]Exciton decay at time at 5 wt% doped film.
[e]Radiative rate constant.
[f]Non-radiative rate constant.

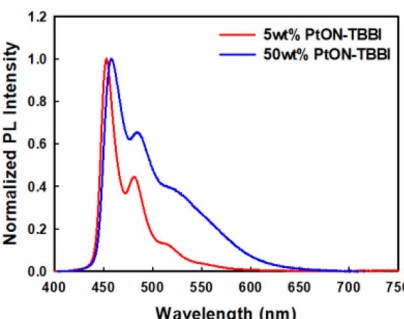
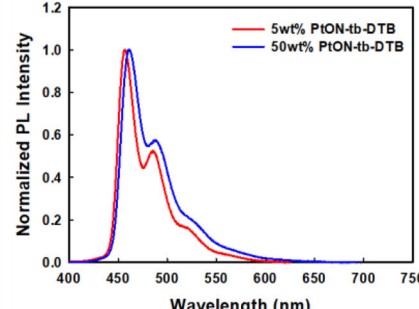
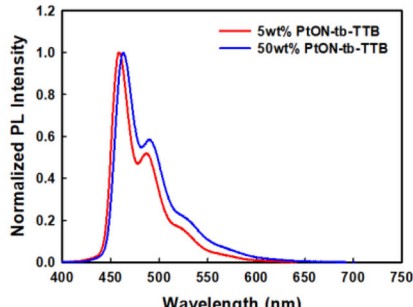

**Fig. 3 | The MMLCT formation characterized by the PL spectra.** 5 and 50 wt% of PtON-TBBI, PtON-tb-DTB, and PtON-tb-TTB doped film in the PMMA matrix (x wt% dopant: (100-x) wt% PMMA matrix).

and rotational motion compared to PtON-TBBI[18]. PtON-tb-TTB has the highest PLQY of 0.99 owing to the higher activation energy (~ 0.842 eV). The calculated values of $k_r^T/k_{nr}^T$ for PtON-TBBI, PtON-tb-DTB, and PtON-tb-TTB are $3.60 \times 10^5/1.89 \times 10^4$, $2.40 \times 10^5/6.77 \times 10^4$, and $3.33 \times 10^5/3.37 \times 10^3$, respectively. PtON-TBBI and PtON-tb-TTB show similar $k_r^T$ values, however, $k_{nr}^T$ values of PtON-tb-TTB are smaller because of its higher activation energy. In addition, the higher $E_{a,T1 \to 3MC}$ of PtON-tb-TTB has suppressed the transition from $T_1$ to 3MC. Further PtON-tb-DTB compared to PtON-tb-TTB has a lower $k_r^T$ value due to decreased molecular rigidity. All the measured photo-physical properties are summarized in Table 2.

Further, investigating the formation of MMLCT, the PL spectra of PMMA films doped with 5 wt% and 50 wt% of the materials were measured as shown in Fig. 3. The differences in the second vibronic peak intensity between the 5 wt% and 50 wt% doped films of PtON-TBBI, PtON-tb-DTB, and PtON-tb-TTB are 0.211, 0.054, and 0.065, respectively. As we expected from QC and MD simulation, the MMLCT formation of PtON-tb-TTB was significantly suppressed owing to the additional t-butyl group. In addition, PtON-tb-DTB shows suppressed MMLCT formation compared with PtON-TBBI, although it has a similar density value in MD simulation. Our results imply that orbital overlap reduction plays a role in MMLCT formation as well. Thus, PtON-tb-DTB has a lower second vibrational peak intensity difference between the 5 wt% and 50 wt% doped films than that of PtON-tb-TTB, and this difference is induced by the higher dihedral angle (∠C4-Pt-N2-C5) of PtON-tb-DTB. This experiment shows that the dihedral angles (∠C4-Pt-N2-C5) can also be crucial parameters to suppress MMLCT formation, besides intermolecular distance.

As mentioned above, MMLCT formation represents orbitals overlapping due to aggregation effects and can cause DET in the film state. Exciton diffusion effects are related to the exciton quenching mechanism. DET and Forster resonance energy transfer (FRET) processes are attributed to exciton diffusion. When diffused excitons encounter each other within the capture radius, which is typically

assumed to be the Van der Waals interaction distance, they are quenched through both FRET and DET mechanisms[28,29]. The energy transfer equations for both processes are as follows[36]:

$$\text{Forster Resonance Energy Transfer Rate} : k_{ET}^F$$
$$= \frac{1}{\tau_D}\left(\frac{R_0}{R}\right)^6, R_0{}^6 = \frac{9000(\ln 10)\Phi_p \kappa_p{}^2}{N_A 128\pi^5 n_D{}^4}J(\lambda) \quad (2)$$

$$\text{Dexter Energy Transfer Rate} : k_{ET}^D = KJ_T e^{-\frac{2R}{L}} \quad (3)$$

Where $k_{ET}^F$ is energy transfer rate through FRET, $k_{ET}^D$ is energy transfer rate through DET, $\tau_D$ is exciton lifetime of energy donor, $R_0$ is the distance at which energy transfer probability is 50%, $N_A$ is Avogadro number, $\Phi_p$ is PLQY, $\kappa_p{}^2$ is dipole orientation ($0.845\sqrt{2/3}$), $n_D$ is refractive index (1.8), $J(\lambda)$ is spectral overlap, K is the specific orbital interaction, $J_T$ is normalized spectrum overlap integral, L is effective electron tunneling distance, which means average Bohr radius between the molecules, and R is the intermolecular distance. To study concentration quenching related to exciton diffusion effects, PMMA films doped with 5, 10, 15, 25, 40, and 50 wt% of PtON-TBBI, PtON-tb-DTB, and PtON-tb-TTB were prepared. The TRPL measurements and lifetime fitting data results for these materials are shown in Fig. 4a and Supplementary Fig. 23. To avoid bimolecular exciton quenching induced at high exciton concentrations, a 340 nm excitation wavelength is selected, because it has a relatively low absorption peak below 380 nm of wavelength, where the emission peak does not mix with the excitation wavelength. Also, it is assumed that intermolecular interaction is limited to a 5 wt% doping level. Figure 4a and Supplementary Fig. 23 reveal that the exciton lifetime decay is mono-exponential[37,38]. A combination of FRET and DET activities disrupts the radiative rate by transferring energy between molecules, resulting in the concentration quenching rate. As a result, the concentration quenching model is represented by Eq. (4), and the

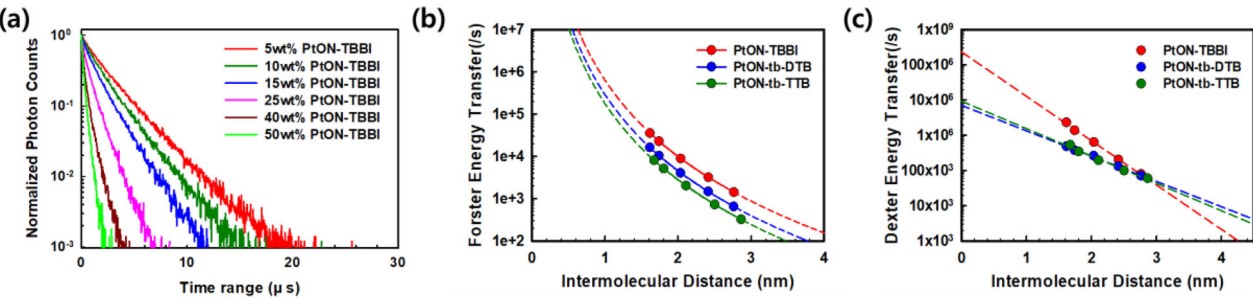

**Fig. 4 | TRPL decay curve and fitted plot according to energy transfer equation. a** TRPL measurements on films as a function of the doping concentration of PtON-TBBI in the PMMA matrix. **b** Forster resonance energy transfer - intermolecular distance. **c** Dexter energy transfer - intermolecular distance.

intermolecular distance is obtained by Eq. (5):

$$\frac{dT}{dt} = -\left(\frac{1}{\tau_{5wt\%}} + k_{ET}^F + k_{ET}^D\right)T \qquad (4)$$

$$R = 2 \times \sqrt[3]{\frac{3M_W}{4\pi\rho\beta N_A}} \qquad (5)$$

Where t, T, $\tau_{5wt\%}$, $M_W$, $\rho$, $\beta$, and $N_A$ are time, triplet exciton density, exciton decay rate at 5 wt% dopped film, molecular weights, film density calculated from MD simulation, doping wt%, and Avogadro number respectively. FRET rates are calculated by using $R_0$ and R. Calculated $R_0$ values of PtON-TBBI, PtON-tb-DTB, and PtON-tb-TTB are 1.09, 0.98, and 0.90 nm, respectively. Calculated FRET rates ($k_{ET}^F = \frac{1}{\tau_{5wt\%}}\left(\frac{R_0}{R}\right)^6$) are significantly lower than DET rates. In Fig. 4b, c, FRET rates are $10^3$-$10^4$ order, whereas DET rates are $10^5$-$10^6$ order. Consequently, the DET mechanism dominates exciton diffusion. Supplementary Tables 1 and 2 summarize the measured exciton lifetime as well as the calculated FRET and DET rates.

To explore DET in the systems under consideration, Eq. (3) is used to fit the DET rate in Fig. 4c. The obtained L and $KJ_T$ values of PtON-TBBI, PtON-tb-DTB, and PtON-tb-TTB are 0.69 nm and $2.36\times10^8$/s, 1.22 nm and $6.78\times10^6$/s, and 1.12 nm and $9.34\times10^6$/s, respectively. The effective tunneling distance (L), which is sum of average Bohr radius for donor and acceptor molecule, means tunneling effects occurs significantly. The additional *t*-butyl groups in PtON-tb-DTB and PtON-tb-TTB attributed to increased L than PtON-TBBI. However, in comparison between PtON-tb-DTB and PtON-tb-TTB, in PtON-tb-DTB has larger L, which is induced from dihedral angle (∠C4-Pt-N2-C5) difference. Whereas, the PtON-tb-TTB complex has a small L value, which is attributed to severe MMLCT formation. In addition to L, $KJ_T$ values show similar tendency with MMLCT formation. Specific orbital interaction dominates $KJ_T$ due to the small $J_T$. High $KJ_T$ indicates strong specific orbital interaction. Thus, the results reveal that DET is dependent on MMLCT formation.

In general, MMLCT exhibits excimer characteristics resulting in a red-shifted and broad spectrum. Additional energy state formation of excimer, which results in bimolecular interaction like excimer, can be affected by the radiative process, leading to changes in $k_r^T$. To investigate changes of $k_r^T$, the PLQY of the films was measured, and a simple equation ($\frac{k_r^T}{k_r^T + k_{nr}^T + k_{ET}^F + k_{ET}^D}$) was used to fit the results. The experimental values are in good agreement with the expected values, as shown in Supplementary Fig. 24. As a function of doping concentration, it implies that triplet exciton diffusion effects over the DET mechanism, rather than variations of $k_r^T$ due to the MMLCT excimer behavior account for the lowered PLQY. Triplet exciton diffusion, associated with specific orbital interaction, perturbs the radiative decay process through DET. Diffusion of triplet excitons is therefore caused via

orbital interaction, which is related to MMLCT formation. In Fig. 4 and Supplementary Fig. 24, experimental values are well correlated with our expected values. It means that bimolecular quenching is sufficiently suppressed in this experiment and exciton quenching occurs through FRET and DET process.

## Electroluminescence (EL) device

To evaluate the device characteristics of the new Pt(II) emitters, phosphorescent OLEDs were fabricated. The optimized device configuration is as follows: ITO (50 nm)/HATCN (7 nm)/PCBBiF (45 nm)/SiCzCz (10 nm)/53 wt% SiCzCz: 35 wt% SiTrzCz: 12 wt% Pt(II)dopant (40 nm)/mSiTrz (5 nm)/mSiTrz: Liq (2:8) (35 nm)/LiF (1.5 nm)/Al (100 nm). Where dipyrazino[2,3-f:2′,3′-h]quinoxaline-2,3,6,7,10,11-hexa carbonitrile (HATCN), N-([1,1′-biphenyl]−4-yl)−9,9-dimethyl-N-(4-(9-phenyl-9H-carbazol-3-yl)phenyl)−9H-fluoren-2-amine (PCBBiF) were used as hole injection and transporting layer[39], respectively 9-(3-(triphenylsilyl)phenyl)−9H−3,9′-bicarbazole (SiCzCz) was used as electron blocking layer and hole transporting p-type host in emissive layer. 9,9′-(6-(3-(Triphenylsilyl)phenyl)−1,3,5-triazine-2,4-diyl)bis(9H-carbazole) (SiTrzCz) was used for electron transporting n-type host. 2-phenyl-4,6-bis(3-(triphenylsilyl)phenyl)−1,3,5-triazine (mSiTrz) was used as hole blocking layer and electron transporting layer in combination with 8-quinolinolato lithium (Liq). Lithium fluoride (LiF) was utilized as electron injection layer. PtON-TBBI, PtON-tb-DTB, and PtON-tb-TTB were introduced as Pt(II) blue phosphorescent dopants. The molecular structures and energy diagrams are presented in Supplementary Fig. 25.

The device characteristics of our novel materials were evaluated and are shown in Fig. 5 and the data were summarized in Table 3. Current density (J)- voltage (V) characteristics were found to be unaffected by the dopant materials as shown in Fig. 5a. It was demonstrated that the employed dopant materials have almost no impact on the parameters of current density (J) versus voltage (V). Compared to Pt(II) dopants, SiTrzCz has a deeper LUMO energy level; hence, it is mostly employed to transport electrons. The Pt(II) dopant exhibited similar hole-trap properties, as they have similar HOMO energy levels. The maximum difference in HOMO energy level between PtON-TBBI and PtON-tb-DTB is only 0.05 eV, which is negligible. Figure 5c shows J-EQE curves of Pt(II) blue phosphorescent OLEDs, with the highest maximum EQE of 26.7% observed for PtON-tb-TTB. However, the EQE of PtON-tb-DTB device was comparatively lower than that of PtON-TBBI. Nevertheless, it showed significantly improved roll-off characteristics. The roll-off characteristics were compared using $J_0$ values, which represent the current density at which the EQE drops to half of the maximum EQE. The $J_0$ values for PtON-TBBI, PtON-tb-DTB, and PtON-tb-TTB were 185, 235, and 203 mA/$cm^2$, respectively. Among them, PtON-tb-DTB demonstrated better roll-off characteristics. Also, non-linear luminescence (L)-J curve, which clarify roll-off characteristics is presented on the Supplementary Fig. 28[40]. Luminescence decline rate of PtON-TBBI, PtON-tb-DTB, and PtON-tb-TTB was calculated at J = 100 mA/$cm^2$

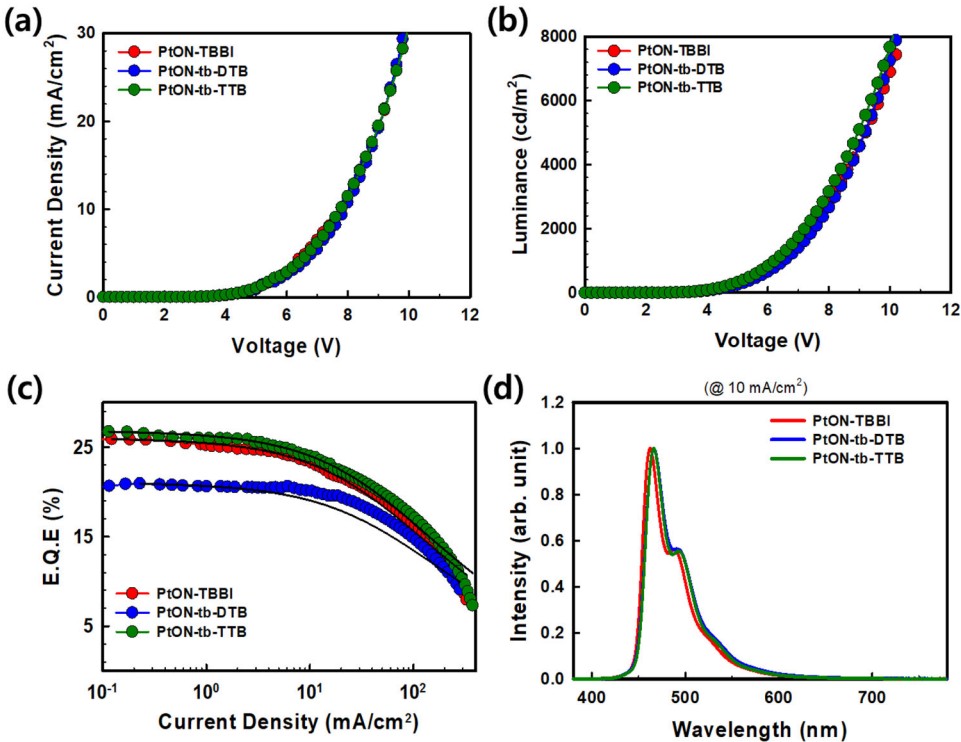

**Fig. 5 | Device characteristics. a** Current density (J) – voltage (V) graph, **b** Luminescence (L) -voltage (V), **c** EQE – current density (J) graph fitted by roll-off model. **d** Electroluminescence spectra (at 10 mA/cm²) of PtON-TBBI, PtON-tb-DTB, and PtON-tb-TTB.

### Table 3 | Summary of PhOLEDs device data for Pt(II) complexes

| | $V_{on}/V_D{}^a$ (V) | $\lambda_{EL}{}^b$ (nm) | $EQE_{max}/EQE_{1000nit}{}^c$ (%) | $J_0{}^d$ (mA/cm²) | LT$_{95}{}^e$ (h) | LT$_{95}{}^f$ (h) | CIE (x,y)$^g$ |
|---|---|---|---|---|---|---|---|
| PtON-TBBI | 2.5/6.3 | 462 | 25.9/24.6 | 185 | 20.2 | 28.0 | (0.14, 0.19) |
| PtON-tb-DTB | 2.5/6.3 | 466 | 20.9/20.4 | 235 | 169.3 | 235.1 | (0.14, 0.22) |
| PtON-tb-TTB | 2.5/6.3 | 466 | 26.7/25.5 | 203 | 31.0 | 43.0 | (0.14, 0.22) |

$^a V_{on}$ and $V_D$ is turn-on (at 1 nit) and driving (at 1000 nit) voltage, respectively.
$^b$Maximum emission wavelength in EL spectrum.
$^c$Maximum EQE and EQE at 1000 nit.
$^d$Critical current density.
$^e$Device lifetime (LT$_{95}$) at 1200 nit.
$^f$Calculated device lifetime (LT$_{95}$) at 1000 nit by using acceleration factor($n = 1.8$).
$^g$CIE color coordinate at 10 mA/cm².

through $\frac{L(ideal)-L(exp)}{L(ideal)} \times 100(\%)$. The calculated values of PtON-TBBI, PtON-tb-DTB, and PtON-tb-TTB are 44.4%, 38.7%, and 44.2%, respectively. A similar tendency was observed in the EQE-J graph.

By measuring PLQYs and TRPL, the variance in maximum EQE was examined. The film PLQY was measured using a 12 wt% dopant mixture of SiCzCz and SiTrzCz host materials. The PtON-TBBI, PtON-tb-DTB, and PtON-tb-TTB dopants had PLQY values of 84%, 73%, and 90%, respectively. Therefore, the differences in maximum EQE were attributed to the PLQY values, which are related to the $E_a$ and non-radiative vibrational process. The TRPL measurement was used to measure the exciton lifetime. The exciton lifetime of PtON-TBBI, PtON-tb-DTB, and PtON-tb-TTB were measured to be 1.98, 1.98, and 2.04 µs, as shown in Fig. 6a, respectively. By using the exciton lifetime and PLQY values, $k_r^T$ and $k_{nr}^T$ were calculated through Eq. (1). The $k_r^T/k_{nr}^T$ values for PtON-TBBI, PtON-tb-DTB, and PtON-tb-TTB were $4.24 \times 10^5/8.08 \times 10^4$, $3.69 \times 10^5/1.36 \times 10^5$, and $4.41 \times 10^5/4.90 \times 10^4$, respectively. The PMMA-doped film showed similar trends. PtON-TBBI and PtON-tb-TTB are quite similar $k_r^T$ values. However, the $k_{nr}^T$ value of PtON-tb-TTB was lower than that of PtON-TBBI. The vibrational relaxation affected by $k_r^T$ value of PtON-tb-DTB to be

lower than that of PtON-TBBI, although they had similar $E_{a,T1 \to 3MC}$ values.

Figure 6b illustrates the device lifetime as recorded at 1200 nit initial luminescence. PtON-TBBI, PtON-tb-DTB, and PtON-tb-TTB had measured lifetimes (LT$_{95}$) of 20.2, 169.3, and 31 h, respectively. The earlier report of the PtON-TBBI device showed the lifetime (LT$_{95}$) of the is about 150 h at 1000 nit[33]. In our experiment, we discovered that PtON-TBBI had a 20.2 h lifetime (LT$_{95}$) at 1200 nit initial luminescence. To compare the device lifetime of the PtON-TBBI device with earlier reports, an acceleration factor of 1.8 was used to consider widely reported values[41–43]. Although 1.8 of the acceleration factors is utilized, a lifetime of PtON-TTBI at 1000 nit is estimated to take about 28.0 h. Despite having nearly identical driving voltage, efficiency, and color coordinates to those earlier reports, the PtON-TBBI device has a substantially different device lifetime[33]. This divergence may originate from different experimental environments and evaporation equipment. In contrast, the PtON-tb-DTB device displayed 169.3 h while being estimated to be 235.1 h at 1000 nit under the same condition. The longevity of PtON-tb-DTB is approximately 8.4 times longer than that of PtON-TBBI. Device lifetime is influenced by TTA and TPA

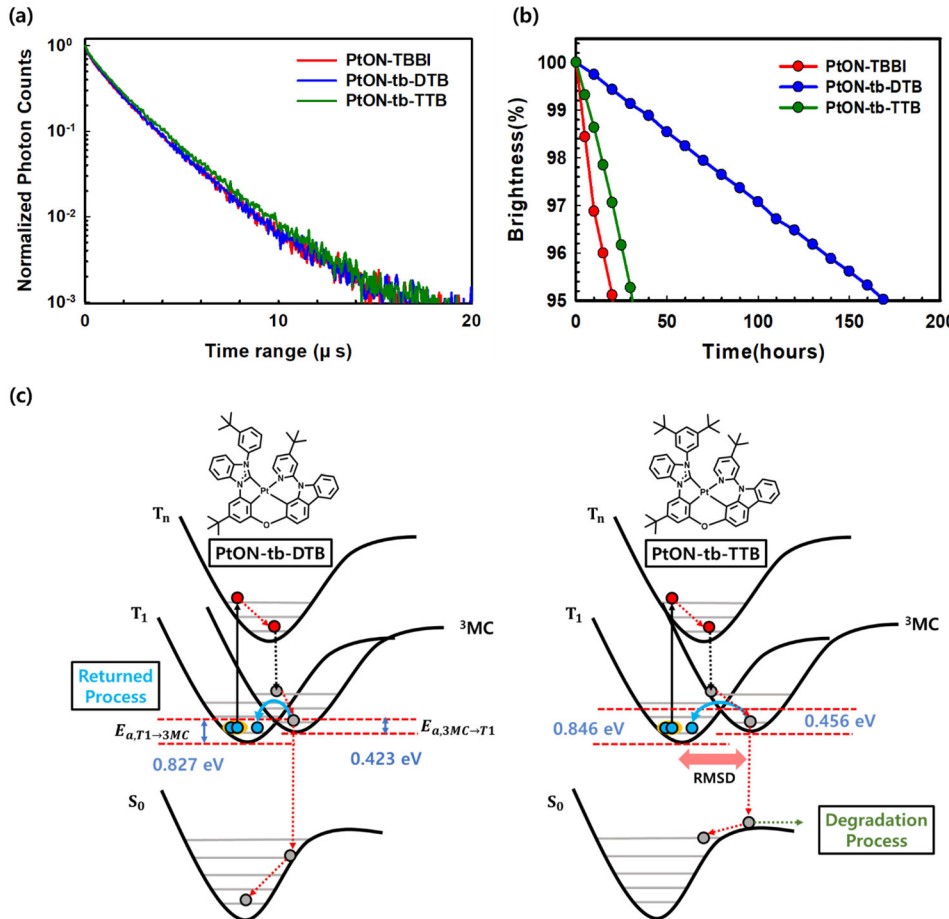

**Fig. 6 | Degradation mechanism. a** TRPL measurements for 12 wt% doped films on SiCzCz and SiTrzCz mixed host. **b** Device lifetime of PtON-TBBI, PtON-tb-DTB, and PtON-tb-TTB at 1200 nit. **c** Schematic diagram of degradation mechanism based on RMSD simulation.

quenching, which are related to roll-off characteristics. To investigate roll-off characteristics, the following roll-off model is introduced[44]:

$$\gamma = \frac{q(\mu_n + \mu_p)}{\varepsilon\varepsilon_0} \tag{6}$$

$$\frac{dP}{dt} = \frac{J}{qd} - \gamma P^2 \tag{7}$$

$$\frac{dT}{dt} = -\frac{1}{\tau_D}T - \frac{1}{2}k_{TT}T^2 - k_{TP}TP + \gamma P^2 \tag{8}$$

Where $\gamma, \mu_n, \mu_p, P, d, k_{TT}, k_{TP}$ are Langevin recombination rate, electron mobility, hole mobility, polaron density, recombination zone thickness, TTA and TPA rate, respectively. $\gamma$ was calculated through Capacitance-Voltage measurements by using half device. Measured average hole and electron mobility and device configurations are presented on Supplementary Fig. 26 and Supplementary Table 3. Thus, calculated $\gamma$ is $5.8 \times 10^{-12} cm^3/s$ at $10^6$ V/cm. $\gamma$ values are similar because of similar J–V characteristics as presented on Fig. 5a. and recombination zone thickness is assumed as EML thickness (40 nm). The $k_{TT}$ and $k_{TP}$ are obtained by fitting the values on Fig. 5d. Calculated $k_{TT}$ and $k_{TP}$ of PtON-TBBI, PtON-tb-DTB, and PtON-tb-TTB were $3.5 \times 10^{-12}/$ $2.9 \times 10^{-14}$, $2.7 \times 10^{-12}/$ $2.9 \times 10^{-14}$, and $3.0 \times 10^{-12}/$ $2.9 \times 10^{-14}$ $cm^3/s$, respectively. Through this roll-off model, it is understandable that TTA mechanism-related hot exciton is a key factor for determining the device's lifetime. The higher $k_{TT}$ devices result in a

shorter device lifetime. Those $k_{TT}$ values are related with exciton diffusion. Diffusivity of materials can be calculated through the formula $k_{TT} = 8\pi R_c D$ where $R_c$ is the capture radius and D is the diffusivity of the triplet[28,45,46]. The $R_c$ values are extracted through intermolecular distance in the film. D value of PtON-TBBI, PtON-tb-DTB, and PtON-tb-TTB is $5.36 \times 10^{-7}$, $4.13 \times 10^{-7}$, and $4.42 \times 10^{-7}$ $cm^2/s$. Thus, D values are well correlated with our MMLCT experiments. Through this analysis, it is clarified that MMLCT formation is related with $k_{TT}$.

Hot exciton (~5 eV) originated from TTA mechanism can overcome the $E_{a,T1\rightarrow 3MC}$ barrier, which can form 3MC state. Ruptured Pt-N bond at 3MC state induces non-radiative decay process[33,47]. Formed 3MC state can transit to $T_1$ by overcoming the $E_{a,3MC\rightarrow T1}$ or the remained 3MC state can be degraded after relaxation process as clearly presented in Fig. 6c. The degradation mechanism is attributed to geometrical changes between 3MC and $T_1$ states. To clarify the conformational changes between them, the root means square displacement distance (RMSD) between 3MC and $T_1$ were calculated. The obtained RMSD of PtON-TBBI, PtON-tb-DTB, and PtON-tb-TTB is 2.53, 2.21, and 2.46 Å, respectively. PtON-tb-DTB has less change in the transition states, even though most of the 3MC exciton is able to transit to the $T_1$ state, instead of which is moved to the ground state. However, PtON-TBBI shows significant conformational changes that can trigger the degrading of the molecules. In addition, PtON-tb-DTB has a lower $E_{a,3MC\rightarrow T1}$ than PtON-tb-TTB. It indicates that the 3MC exciton population, which induces molecular degradation, can be reduced significantly. Further, to clarify material stability from TTA, a photo-stability test is conducted using UV-LED, which emits 360 nm of wavelength, and 12 wt% doped film on the DPEPO

host in Supplementary Fig. 27. A similar tendency was observed, PL intensity change at maximum emission wavelength of PtON-TBBI, PtON-tb-DTB, and PtON-tb-TTB doped film show 4%, 19%, and 31% reduction of degradation for 12 h. Based on these findings, for a stable blue phosphorescent dopant Pt(II) complex, the 3MC exciton formation should reduce, and geometrical changes occur between $T_1$ and 3MC states, which is a key factor. 3MC exciton can be generated by overcoming $E_{a,T1 \to 3MC}$ through various mechanisms. One of the important mechanisms to overcome $E_{a,T1 \to 3MC}$ is hot exciton formation through TTA, which is relatively easy with strong MMLCT formation. It should be drastically suppressed to form MMLCT. In addition, to reduce 3MC exciton population, $E_{a,3MC \to T1}$ can be considered as one of the important parameters and lower $E_{a,3MC \to T1}$ facilitates the revert process from 3MC to $T_1$ state. Through this promising concept the 3MC exciton population can be reduced significantly. As a result, the device's lifetime can be extended.

## Discussion

In summary, two tetradentate Pt(II) complex materials, PtON-tb-DTB and PtON-tb-TTB, have been synthesized using ligand manipulation as blue phosphorescent dopants. The addition or removal of bulky *t*-butyl groups on the ancillary ligands has a significant impact on the suppression of MMLCT formations. The *t*-butyl substitution position affects the intermolecular distance between dopants and the dihedral angle (C4-Pt-N2-C5) at the excited state, resulting in reduced MMLCT formation by suppressing orbital overlap. Thus, exciton diffusion caused by the DET process is suppressed. Devices utilizing these dopants have achieved high EQEs of 26.3%, 20.9%, and 25.9% for PtON-tb-DTB, PtON-TTB, and PtON-TBBI, respectively. Although the lowest EQE of PtON-tb-DTB originates from the lowest PLQY of 73% in 12 wt% doped films, the PtON-tb-DTB device has demonstrated exceptional operational stability with a lifetime of 169.3 h at an initial luminescence of 1200 nits, which is 8.4 times longer than that of the PtON-TBBI complex. The device utilizing PtON-tb-DTB exhibits superior roll-off characteristics, and the device lifetime is proportional to $J_O$ values. Thus, the lifetime difference among dopants is analyzed through a roll-off analytic model, which is related to TTA and TPA. As a result of roll-off analysis, the roll-off difference according to dopant comes from $k_{TT}$, which is related to exciton diffusion, rather than $k_{TP}$. Also, it was clarified through a UV stability experiment. The $T_1$ state has transited to the 3MC state through hot exciton formation caused by TTA. The formed 3MC state can return to the $T_1$ state or lead to molecular degradation after the transition to the ground state because of the large conformational change between 3MC and the $T_1$ state. Therefore, suppression of MMLCT can reduce the TTA process by reducing exciton diffusion in the device, and lower $E_{a,3MC \to T1}$ facilitates conversion rapidly from the 3MC to the $T_1$ state. Both effects are attributed to a reduction in the 3MC exciton population. In addition, there were tiny geometrical changes between 3MC and $T_1$, which enhanced the molecular stability. Our design approach and analysis results have important implications for the future development of blue phosphorescent Pt(II) complexes.

## Methods
### Materials
All the reagents and solvents were purchased from commercial suppliers, including Aldrich Inc., Tokyo Chemical Industry Co., Ltd. (TCI), and Alfa Aear. Dichloro(1,5-cyclooctadiene) platinum (II) was generously provided by Furuya Metal Korea. HATCN and Liq were purchased from the EM Index and Osccila, respectively. PCBBiF, SiCzCz, SiTrzCz, and mSiTrz were synthesized by previously reported procedures[33,39].

### Characterization
All the chemical reactions and characterization were performed under a nitrogen atmosphere. Proton nuclear magnetic resonance ($^1$H NMR) and carbon nuclear magnetic resonance ($^{13}$C NMR) spectra were recorded on a Bruker DRX 300 MHz spectrometer. High-resolution mass spectra (HRMS) were examined by quadrupole time of flight (Q-Tof-MS) methods with a Xevo G2-XS Tof. To predict the photo-physical properties of the materials, we used methylene chloride solution at a concentration of $1 \times 10^{-5}$ M. The UV-vis absorption spectrum was measured by the V-750 Spectrophotometer (Jasco), and the solution PL and low temperature (77 K) PL spectra were measured by the FP-8500 Spectrofluorometer (Jasco). The total (or absolute) PLQY of doped films was measured using an integrating sphere system under an inert atmosphere. Transient PL decay measurements were recorded in both doped film and solution in an inert atmosphere, using the Quantaurus-Tau fluorescence lifetime measurement system (C11367-03, Hamamatsu Photonics Co). Electrochemical properties of dopant materials were measured by (EC epsilon electrochemical analysis equipment) cyclic voltammetry (CV) using the materials coated on a 50 nm thickness of ITO/glass substrate as a working electrode, platinum wire, carbon wire, and Ag wire with 0.01 M AgNO$_3$ (a counter, working, and reference electrode, respectively). Tetrabutylammonium perchlorate (Bu$_4$NClO$_4$) 0.1 M was used as a supporting electrolyte in an acetonitrile solution. The potential values were converted to the saturated calomel electrode (SCE) scale using an internal ferrocene/ferrocerium (Fc/Fc$^+$) standard. The thermal stability was measured by TA 2050 TGA, a thermogravimetric analyzer (TGA) with the sample heated at a rate of 10 °C/min. Differential Scanning Calorimetry analysis (DSC) was done using TA Instruments 2100 DSC, with the sample heated at a rate of 10 °C/min from 0 °C to 300 °C under an inert atmosphere.

### Device fabrication and performance measurements
For the fabrication of OLEDs, ITO-coated glass substrates with a sheet resistance of 10 $\Omega$/m$^2$ and a thickness of 50 nm were used. The substrate was cleaned prior to deposition by ultrasonic treatment with acetone and isopropyl alcohol, followed by cleaning with deionized water and drying under nitrogen. The cleaned substrates were further treated with UV-ozone for 10 min. All organic layers and the metal cathode were deposited on the cleaned ITO/glass substrates using a vacuum evaporation system (under the vacuum pressure of ~1×10$^{-7}$ Torr and a deposition rate of approximately 0.5 Å/s). The deposition rates of lithium fluoride (LiF) and aluminum (Al) were maintained at 0.1 and 4.0 Å/s, respectively. Film deposition and encapsulation process were used to inhibit the degradation in the nitrogen-filled glove box. The OLEDs had a uniform area of 4 mm$^2$ for all the samples studied in this work. The J–V and L–V characteristics of the fabricated devices were measured using a Keithley 2635 A SMU and Konica Minolta CS-100A, respectively. EL spectra and CIE 1931 color coordinates were observed with a Konica Minolta CS-2000 spectroradiometer. All measurements were conducted under ambient conditions.

### Theoretical simulation
QC simulations for the lowest triplet state ($T_1$), metal centered triplet (3MC), and transition state were performed using density functional theory (DFT) calculations by B3LYP-D3 functional and LACV3P + +** basis set. A transition state search was conducted using Quadratic Synchronous Transit (QST) search methods. The calculations were done by Schrödinger Materials Science 4.6 suite and were implemented using Jaguar Quantum Chemical Engine (Wallingford, CT, USA). MD simulations using the Desmond MD engine and Schrödinger Material Science 4.6 software. In the disordered system, 500 molecules exist. The material relaxation strategy that used 20 ps NVT Brownian minimization at 10 K, 20 ps NPT Brownian minimization at 100 K, and a 100 ps NPT MD stage at 300 K was used to stabilize the disordered system.

## Data availability
The authors declare that the data supporting the findings of this study are available within the paper and its supplementary information files.

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

## Acknowledgements

This work was funded by The Package Program of Materials and Parts (20015850), the Establishment of Display Innovation Process Platform (20020408, 20006464) funded by the Ministry of Trade, Industry & Energy (MOTIE, Korea), and Program of the National Research Foundation of Korea (NRF) grant funded by the Ministry of Education (RS-2023-00301974). This work was funded by Samsung Display.

## Author contributions

Y.H.J. carried out analysis of device and wrote the manuscript, G.S.L. carried out the synthesis and characterization of platinum complex, S.M. carried out the synthesis of the organic material to fabricate device and supported manuscript wrote. H.R.K. and J.H.O. carried out fabrication of devices. J.H.H. measured and studied photo-physical property of organic material. S.B.Y. carried out synthesis of the organic material to fabricate device. J.H.L. supported synthesis of platinum complex, M.Y.C. supported manuscript writing. All the experiments and characterization were conducted under the supervision of Y.H.K. and J.H.K.

## Competing interests

The authors declare no competing interests.
