## [Peer Review File · Nature Communications]

Modified t-butyl in tetradentate platinum(II) complexes enables exceptional lifetime for blue-phosphorescent organic light-emitting diodesREVIEWER COMMENTS

Reviewer #1 (Remarks to the Author):

Please see the attached file. Thanks.

Comments:

The highly efficient phosphorescent blue OLEDs typically suffer from their relatively short operational lifetimes, which impedes their commercial application in electronics. In this work, Kwon, Kim and co-workers designed two new tetradentate NHC-based Pt(II) complexes, which had high PLQYs of 99% and 78% in doped films, respectively. PtON-tb-TTB and PtON-tb-DTB doped blue OLEDs demonstrated peak EQEs of 26.3% and 20.9%, respectively, and the PtON-tb-DTB-based device exhibited long operational lifetime of 169.3 h at L_0 of 1200 cd/m². The authors investigated the factors on the device stability and degradation mechanisms of the blue OLEDs. This work should provide a very valuable reference for the further development of efficient and stable phosphorescent Pt(II) complexes for blue OLEDs. Therefore, I recommend it to publish in *Nature Communications* after the below comments addressed, which might be helpful to the authors to further improve the quality of the manuscript.

1. In the second paragraph of the "Introduction" section, the authors presented the progress of the operational lifetimes of blue TADF and phosphorescent OLEDs, and listed the lifespan values of each device. However, the operational lifetimes are greatly related to the color purity of the blue OLEDs (CIE_y values). Therefore, please add the CIE_y values to avoid the misunderstandings caused by the only lifespan values. Please also provide the CIE values of the newly fabricated blue OLEDs in Table 3.

2. Lines 52–54, "Indeed, blue TADF OLEDs have insufficient device stability compared to phosphorescent OLEDs because of their high triplet energy, which is induced by the emission characteristics of TADF." This statement confuses me. If the blue TADF OLEDs and phosphorescent OLEDs have the same dominant emission peaks, the triplet energy of TADF emitters should be lower than those of the phosphorescent emitters, because the TADF is from T₁→S₁→S₀, but phosphorescence is directly from T₁ to S₀.

3. Lines 67–71, "In pioneering reports, J. Li et al, PtON1 and PtON7 were developed and examined by using tetradentate cyclometalated ligand and phenyl methylimidazole or phenyl pyrazole ancillary ligands. **These blue Pt(II) complexes** drastically deteriorated PL spectra because of their ancillary ligand^[31, 32]. Further, **this issue** can be resolved by introducing a bulky substitution on the pyridine unit of the primary NHC ligand. Thus, much research was done by using different electron-rich bulky substitutions such as alkyls, long chain, adamantly, aromatic, and heterocyclic groups, respectively." This statement was confused. First, what does "These blue Pt(II) complexes..." refer to? The PtON1 or PtON7 from J. Li's work, or the complexes in ref. [31, 32]? They are the Pt(II) complexes from different literatures. Second, does "this issue" mean the broad PL spectra? If so, the issue can be resolved not because of the introduction of a bulky substitution, but because of the electron-donating property of the substitution at the *para*-position of pyridine (*ortho*-position or *meta*-position does not work), which can increase the ³MLCT level, and make the excited-state properties of the Pt(II) complexes possess ³LC (or ³LE) dominated emission with some ³MLCT character, thus, realize narrow emission spectra [*Inorg. Chem.* **2017**, *56*, 8244; *Inorg. Chem.* **2019**, *58*, 12348; *Inorg. Chem.* **2020**, *59*, 13502 (Figure 10)]. Third, introduction of aromatic, and heterocyclic groups (like carbazole) to the 4-position of pyridine can not enable the emission spectra to become narrow.

4. Lines 76–78, "In addition, the *t*-butyl group substituted on the meta position of the ether linkage phenyl group on PtON7-dtb which is attributed to increased intermolecular distance and obtained negligible shoulder peak with narrow PL spectra." Actually, just as the analysis in above comment 3, the introduction of electron-donating property of the substitution to the 4-position of pyridine can increase the ³MLCT level, and realize narrow emission spectra [*Inorg. Chem.* **2017**, *56*, 8244]. It is true that the *t*-butyl group substituted on the meta position of the ether linkage phenyl group on PtON7-dtb could increase intermolecular distance, however, it

could also slightly increase the height of the shoulder peak (PtON7-dtb vs PtON7-tbu) [*Inorg. Chem.* **2017**, *56*, 8244 (Figure 8)], which is unfavorable to the development of narrow PL spectra.

5. In Figure 1, about the *meta*-tBu-phenyl group on the NHC moiety, the tBu should have an effect on the molecular geometry of PtON-tb-DTB, in particular, on the **Transition State** of the PtON-tb-DTB (the steric hindrance between two tert butyl groups appears to be significant in the current figure of the **Transition State**). It is more likely that the spatial steric hindrance of the **Transition State** between the *meta*-tBu of phenyl group on the NHC and the tBu on the Py will be smaller, if the rotate the *meta*-tBu-phenyl group about 180 degrees (or remove the tBu to the other meta position). Please compare the ground state energy levels of the two molecular geometries, and also their influence on the $\angle\text{C-N-C-C}$, $\angle\text{C-Pt-N-N}$ and the **Transition State**.

6. Lines 204–205, “As previously shown, adding the *t*-butyl group at the meta-position of the ether linkage phenyl ring increases conjugation.” In my opinion, the *t*-butyl is an electron-donating group, which can increase the electron density of the HOMO in PtON-tb-DTB compared to PtON-TBBI, this enable the PtON-tb-DTB to be easier to oxidate, thus, PtON-tb-DTB (–5.50 eV) has shallower HOMO level than that of PtON-TBBI (–5.55 eV). Adding the *t*-butyl group at the meta-position of the ether linkage phenyl ring hardly change the molecular geometry of the Ph-Carbene moiety, please carefully consider whether the “increases conjugation” is reasonable?

7. As previous reports (*J. Appl. Phys.* **2000**, *87*, 8049; *J. Am. Chem. Soc.* **2017**, *139*, 9783; et, al.), the OLEDs containing TTA process typically exhibit nonlinear Luminance vs current density (*L–J*) characteristics, therefore, please provide the *L–J* curves of all the three devices in **Figure 5**. This might be a direct experiment evidence for the existence of TTA at the current densities of the device lifetime measurements.

8. Please add the excited lifetime values of the films in **Figure 6**, which can help the readers to check the radiative rates and non-radiative rates of the films. By the way, the non-radiative rate of PtON-tb-TTB (Line 357) should be incorrect.

9. For the PtON-TBBI-based blue OLEDs, the device lifetime LT_{95} was 20.2 h at L_0 of 1200 nit, which was estimated about 28.0 h. This device lifetime was much shorter than that of the previous report with LT_{95} of about 150 h (*Nat. Photonics* **2022**, *16*, 212.) using the same device structure with different thickness of some functional layers. The possible reasons should be discussed in the manuscript. How about the device lifetime using the same thickness of functional layers as the reference (ITO (50 nm)/ HATCN (10 nm)/ PCBBiF (60 nm)/ SiCzCz (5 nm)/ 60 wt% SiCzCz: 27 wt% SiTrzCz: 13 wt% Pt(II)dopant (35 nm)/ mSiTrz (5 nm)/ mSiTrz: Liq (5:5) (35 nm)/ LiF (1.5 nm)/ Al (100 nm))?

10. In Table 3, the acceleration factor ($n = 1.8$) might be overestimated for Pt(II) complexes-based blue OLEDs, it is suggested that the author obtain actual n values through experiments according previous reports (*Adv. Mater.* **2017**, *29*, 16.5002; *Nat. Photonics* **2021**, *15*, 230.). The experiments are not complicated.

11. In Supporting Information, the chemical structures of ligands **2**, **4** and **5** are incorrect, please revise them.

12. For all HRMS data in Supporting Information, the $[\text{M}+\text{H}]^+$ data should be given to correspond to experimental values (found values).

13. Considering the high requirements of this journal, please provide the characterization data of ligands **4** and **5**.

Other minor issues:

14. "N-heterocyclic carbene", "t-butyl",

"N-([1,1'-biphenyl]-4-yl)-9,9-dimethyl-N-(4-(9-phenyl-9H-carbazol-3-yl)phenyl)-9H-fluoren-2-amine", "9-(3-(triphenylsilyl)phenyl)-9H-3,9'-bicarbazole", "9,9'-(6-(3-(Triphenylsilyl)phenyl)-1,3,5-triazine-2,4-diyl)bis(9H-carbazole)",... the letters "N", "t" and "H" should professionally be italics. Please double check the whole manuscript and Supporting Information and revise them.

15. Please revise "LT50", "LT95" ... as "LT₅₀", "LT₉₅" ... in the whole manuscript.

16. Please revise "3MLCT", "3LC", "3MC" ... as "³MLCT", "³LC", "³MC" ... in the whole manuscript, also in Figure 6.

17. Lines 56–58, please revise "Ir" as "Ir(III)", which is consistency with "Pt(II)".

18. Lines 67–68, please add the reference; line 68, revise "and" as "containing".

19. Line 59, "low color purity caused broad spectrum" should be "low color purity caused by broad spectrum"?

20. Line 60, "small" is better than "narrow" for FWHM? Typically, "narrow" is used with "spectrum".

21. Line 87, "EQEs" should be "EQE"?

22. Line 91, "tetradentate-based Pt(II) complexes" should be "tetradentate NHC-based Pt(II) complex" or "tetradentate NHC ligand-based Pt(II) complex"?

23. Lines 94–97, "In this study, we introduce **two** new Pt(II) complexes, **PtON-tb-DTB and PtON-tb-TTB**, ... These complexes exhibited high PLQYs of **78%, 99%, and 95%**, respectively." The statement confuses me, the two complexes are incompatible with three PLQY values, please revise it.

24. It will be clearer if the black background is changed to white in Figure 1.

25. Line 142, "The decreased \angle C-N-C-C of PtON-tb-DTB influenced, increased conjugation length." is difficult to understand.

26. Typically, the "Measurements" section should be placed before the "synthesis" section in Supporting Information.

27. Line 214, "Clearly, the new materials exhibit a longer lifetime than PtON-TBBI." should be "Clearly, the new materials exhibit longer lifetimes than that of PtON-TBBI."?

28. Lines 272 and 274, The "**Figures 4(a) and S17**" should be "**Figures 4(a) and S16**".

29. Line 326, "N-type host" should be "*n*-type host".

30. Line 337, "However, the EQEs of PtON-tb-DTB devices were comparatively lower than that of PtON-TBBI." should be "However, the EQE of PtON-tb-DTB device was comparatively lower than that of PtON-TBBI."?

31. The unit of time in **Figure 2c** should be " μ s", not "us", the same questions are also in **Figure 4a** and **Figure 6a**.

Reviewer #2 (Remarks to the Author):

This article studied the effect of the substitution position of the t-butyl group in reference Pt(II) complexes on EL performances of the fabricated phosphorescent blue OLEDs based on PtON-tb-TTB and PtON-tb-DTB. They gave detailed analysis and it can be found that the substitution position indeed has large effect on the efficiency and lifetime of the fabricated devices. They attributed the large improvement to the decreased TTA process and better hot exciton stability. This work is still very meaningful for further designing and synthesizing high-efficiency and long lifetime blue phosphorescent OLEDs materials. However, as we see, the basic structure of the types of materials has been reported [Nat. Photonics, 16, 212-218 (2022)], where excellent EL efficiency and lifetime have been obtained, although the authors provided a more detailed analysis of material design and influencing factors in this article. Therefore, the article lacks sufficient novelty to be considered for publication in NC. I suggest to submit this article to a more professional magazine. At the same time, the authors should also consider the following issues:

1. "lifespan" is usually written as "lifetime".
2. In abstract, the 169.3 h lifetime should be TL95 at 1200 cd/m², which should be clearly stated.
3. As shown, PtON-tb-DTB emitted the lowest PLQY and EL efficiency, but have the longest lifetime than PtON-TBBI and PtON-tb-TTB, which is usually not easy to understand.
4. The energy level diagrams of PtON-tb-DTB, PtON-TBBI and PtON-tb-TTB are necessary to be given by calculating or measuring.
5. What is the physical basis of Figure 6(c). As we see, the energy level of S0 in PtON-TBBI is completely different from that of PtON-tb-DTB.

Reviewer #3 (Remarks to the Author):

This manuscript reports the approach of introducing tert-butyl group in an appropriate position to suppress MMLCT in platinum(II)-based emitters to achieve high PLQY of up to 99% in doped films and for improving OLED EQEs up to 26.3%. Extensive computational studies, involving MD, QM, were used to calculate intermolecular distance, rate of excited state processes and deactivation processes, and behaviour in the solid state. Even though it is known that increasing the steric bulk of the square planar emitters can suppress intermolecular interaction, this work has demonstrated that the placement of these tert-butyl group must be strategic in controlling the various excited state processes for performance enhancement. I would recommend this work to be published in nature communication provided the following issues are addressed.

1. MS lines 130-131, how does the removal of one tert-butyl group on the benzimidazole carbene and the flexibility created by such modification correlate with the formation of MMLCT?

2. MS Lines 137-138, Can you elaborate more on how the conjugation length is changed by the introduction of the tert-butyl group on the meta position of tether linkage phenyl ring? I don't see much difference in the calculated structure.
3. MS Fig 2b, since the spectra of two complexes are shifted relative to the reference. Would it be better to unlabel the x-axis?
4. MS Fig3 and Lines 247-248, is the peak intensity difference between 5 wt% and 50 wt% doped films for PtON-tb-DTB very significantly different from that of PtON-tb-TTB?
5. MS Lines 272-273, the absorption of these complexes at 340 nm is not really that poor based on the UV spectra. But how exactly can one avoid bimolecular exciton quenching by choosing this wavelength?
6. Please provide procedure for result fitting in the supporting information.

“Point-to-Point” Responses to Reviewers’ comments

Research article No: NCOMMS-23-38862A-Z

Title: “Effects of Substitution and Position of *t*-butyl groups in Tetradentate Platinum(II) complexes enable exceptional **Lifetime** for Blue Phosphorescent Organic Light-Emitting Diodes”

We are grateful for the careful evaluation and constructive comments from the reviewers. We have attached our point-to-point responses. The original comments from the reviewers are in black. The response to the comment is in **blue**. The corresponding changes made in the revised manuscript are in highlighted **yellow color**.

Thank you for your comments and valuable suggestions regarding our work, which helped us to improve our manuscript quality.

Reviewer #1 (Remarks to the Author):

The highly efficient phosphorescent blue OLEDs typically suffer from their relatively short operational lifetimes, which impedes their commercial application in electronics. In this work, Kwon, Kim and co-workers designed two new tetradentate NHC-based Pt(II) complexes, which had high PLQYs of 99% and 78% in doped films, respectively. PtON-tb-TTB and PtON-tb-DTB doped blue OLEDs demonstrated peak EQEs of 26.3% and 20.9%, respectively, and the PtON-tb-DTB-based device exhibited long operational lifetime of 169.3 h at L0 of 1200 cd/m². The authors investigated the factors on the device stability and degradation mechanisms of the blue OLEDs. This work should provide a very valuable reference for the further development of efficient and stable phosphorescent Pt(II) complexes for blue OLEDs. Therefore, I recommend it to publish in Nature Communications after the below comments addressed, which might be helpful to the authors to further improve the quality of the manuscript.

Response: We appreciate the suggestion of the referee to publish this work in Nature Communications, subject to revisions. The comments mentioned by the reviewer have been addressed, as described in the following responses.

1. In the second paragraph of the “Introduction” section, the authors presented the progress of the operational lifetimes of blue TADF and phosphorescent OLEDs, and listed the lifespan values of each device. However, the operational lifetimes are greatly related to the color purity of the blue OLEDs (CIE_y values). Therefore, please add the CIE_y values to avoid the misunderstandings caused by the only lifespan values. Please also provide the CIE values of the newly fabricated blue OLEDs in Table 3.

Response: Thanks for your valuable suggestion and as you commented, the CIE_y values are related to the device's lifetime, as they indicate the sensitivity of the human eye. In our research, we found that PtON-TBBI has the lowest CIE_y value, and it shows the shortest device lifetime. However, we also observed significant differences in device lifetime among the three materials tested, even though the CIE_y values of PtON-tb-DTB and PtON-tb-TTB were slightly higher than those of PtON-TBBI. Nevertheless, our research shows that CIE_y is not significantly affected, and we have provided the CIE coordinates in **Table 3**.

Table 3. Summary of PhOLEDs device data for Pt(II) complexes.

	V_{on}/V_D^a (V)	λ_{EL}^b (nm)	$EQE_{max}/EQE_{1000\ nit}^c$ (%)	J_0^d (mA/cm ²)	LT ₉₅ ^e (h)	LT ₉₅ ^f (h)	CIE (x,y) ^g
--	--------------------	-----------------------	-----------------------------------	-------------------------------	-----------------------------------	-----------------------------------	------------------------

PtON-TBBI	2.5/ 6.3	462	25.9/ 24.6	185	20.2	28.0	(0.14, 0.19)
PtON-tb- DTB	2.5/ 6.3	466	20.9/ 20.4	235	169.3	235.1	(0.14, 0.22)
PtON-tb- TTB	2.5/ 6.3	466	26.7/ 25.5	203	31.0	43.0	(0.14, 0.22)

^a V_{on} and V_D is turn-on (at 1 nit) and driving (at 1000 nit) voltage, respectively. ^b Maximum emission wavelength in EL spectrum. ^c Maximum EQE and EQE at 1000 nit. ^d Critical current density. ^e Device lifetime (LT_{95}) at 1200 nit. ^f Calculated device lifetime (LT_{95}) at 1000 nit by using acceleration factor ($n=1.8$), ^gCIE color coordinates at 10 mA/cm².

2. Lines 52–54, “Indeed, blue TADF OLEDs have insufficient device stability compared to phosphorescent OLEDs because of their high triplet energy, which is induced by the emission characteristics of TADF.” This statement confuses me. If the blue TADF OLEDs and phosphorescent OLEDs have the same dominant emission peaks, the triplet energy of TADF emitters should be lower than those of the phosphorescent emitters, because the TADF is from $T1 \rightarrow S1 \rightarrow S0$, but phosphorescence is directly from $T1$ to $S0$.

Response: We agreed with your statement but usually, TADF materials have strong Intramolecular Charge Transfer (ICT) characteristics, which induce red-shifted emission, Therefore, TADF materials have higher bandgap and triplet energy. For example, CN-Ir and DBA-DI show 459 and 467 nm of maximum emission wavelength in toluene solution, respectively. However, their triplet energy is 2.70 and 2.95 eV, respectively. (Adv. Optical Mater, 8, 11, 2000102 (2020), Adv. Optical Mater, 9, 13, 2100203 (2021)).

To be clear in our description, we have changed our manuscript as follows.

Blue TADF OLEDs have been observed to exhibit poorer device stability in comparison to phosphorescent OLEDs. This is because the TADF material's strong intramolecular charge transfer characteristics lead to red-shifted emission behavior, which in turn results in higher bandgap and triplet energy characteristics. These higher energy values can cause some reduction in the device stability of blue TADF OLEDs. [6, 8, 40]

3. Lines 67–71, “In pioneering reports, J. Li et al, PtON1 and PtON7 were developed and examined by using tetradentate cyclometalated ligand and phenyl methylimidazole or phenyl pyrazole ancillary ligands. These blue Pt(II) complexes drastically deteriorated PL spectra because of their ancillary ligand[31, 32]. Further, this issue can be resolved by introducing a bulky substitution on the pyridine unit of the primary NHC ligand. Thus, much research was done by using different electron-rich bulky substitutions such as alkyls, long chain, adamantly, aromatic, and heterocyclic groups, respectively.” This statement was confused. **First**, what does “**These blue Pt(II) complexes...**” refer to? The PtON1 or PtON7 from J. Li’s work, or the complexes in ref. [31, 32]? They are the Pt(II) complexes from different literatures. **Second**, does “**this issue**” mean the broad PL spectra? If so, the issue can be resolved not because of the introduction of a bulky substitution, but because of the electron-donating property of the substitution at the para-position of pyridine (ortho-position or meta-position does not work), which can increase the 3MLCT level, and make the excited-state properties of the Pt(II) complexes possess 3LC (or 3LE) dominated emission with some 3MLCT character, thus, realize narrow emission spectra [Inorg. Chem. 2017, 56, 8244; Inorg. Chem. 2019, 58, 12348; Inorg. Chem. 2020, 59, 13502 (Figure 10)]. **Third**, introduction of aromatic, and heterocyclic groups (like carbazole) to the 4-position of pyridine can not enable the emission spectra to become narrow.

Response: We appreciate the insightful recommendation from the reviewer and acknowledge his concerns.

- i. For the first question, Yes we apologies, In this part, we have discussed “these blue Pt(II) complexes such as PtON1 and PtON7 from Jian Li et al earlier reports.
- ii. For the second question, “this issue” means the broad spectrum of PtON1 and PtON7. As you commented, bulky substitution cannot be a solution to resolve the broad-spectrum issue. Yes, we

have agreed with your perceptive statement that the substitutions on the *para*-position of pyridine should have to donate properties and suppress the state mixing between 1MLCT/3MLCT and 3LC state.

- iii. For the third question, this question is related to the second question. Selection of substitution on the *para*-position of pyridine is important to suppress the state mixing, although substitution has donating properties. For example, PtON1-Cz and PtON1-Ph show a broader spectrum than PtON1. To suppress the state mixing, alkyl substitutions, methyl, and dimethyl amine type donating type substitution should be introduced on the *para*-position of pyridine.

To be clear in our description, we have changed our manuscript as follows.

*In the previous reports, J. Li et al. examined the spectrum broadening of the substitution position in PtON1 and PtON7, which are composed of ancillary ligands that are either phenyl pyrazole, methylimidazole, or tetradentate cyclometalated ligands. PtON1 and PtON7, without any additional substitution on the ligand's motif, can result in drastically broad PL spectra. Further, this issue has been addressed by introducing alkyl and dimethyl amine donating substitutions on the para-position of the pyridine unit in the primary NHC ligand, and it shows a narrower spectrum, which effectively suppresses the state mixing between ¹MLCT (metal-to-ligand charge transfer), ³MLCT, and ³LC (ligand center) states.^[32] Later, J.J. Kim et al. introduced the adamantyl group on the para-position of pyridine to decrease MMLCT formation, which leads to a narrow spectrum in solution and film states.^[31] However, alkyl substitutions such as methyl and adamantyl groups have an unsuitable application as substituents owing to their insufficient bulkiness or high molecular weight compared to the *t*-butyl group, respectively. On the other hand, J. Li et al. have reported PtON7-*t*Bu, which shows a narrow spectrum with bulky substitution of the *t*-butyl group incorporated on the primary ligand pyridine unit para-position.*

4. Lines 76–78, “In addition, the *t*-butyl group substituted on the meta position of the ether linkage phenyl group on PtON7-dtb which is attributed to increased intermolecular distance and obtained negligible shoulder peak with narrow PL spectra.” Actually, just as the analysis in above comment 3, the introduction of electron-donating property of the substitution to the 4-position of pyridine can increase the 3MLCT level, and realize narrow emission spectra [Inorg. Chem. 2017, 56, 8244]. It is true that the *t*-butyl group substituted on the meta position of the ether linkage phenyl group on PtON7-dtb could increase intermolecular distance, however, it could also slightly increase the height of the shoulder peak (PtON7-dtb vs PtON7-tbu) [Inorg. Chem. 2017, 56, 8244 (Figure 8)], which is unfavorable to the development of narrow PL spectra.

Response: Thanks for your valuable suggestion, and we agree with your statement that the *t*-butyl substitution on the *meta*-position of the ether linkage phenyl group enhances the vibrational peak, which induces the broad spectrum. Nevertheless, *t*-butyl substitution on the *meta*-position of the ether linkage phenyl group can suppress MMLCT formation by increasing intermolecular distance. It observed a narrow spectrum of the film's state. To be clear in our description, we have changed our manuscript as follows:

*In addition, the *t*-butyl group substituted on the meta-position of the ether linkage phenyl group on PtON7-dtb which is attributed to suppressed MMLCT formation. It can improve color purity in the film state, although it shows enhanced shoulder peak in the solution state.*

5. In Figure 1, about the meta-*t*Bu-phenyl group on the NHC moiety, the *t*Bu should have an effect on the molecular geometry of PtON-tb-DTB, in particular, on the Transition State of the PtON-tb-DTB (the steric hindrance between two tert butyl groups appears to be significant in the current figure of the Transition State). It is more likely that the spatial steric hindrance of the Transition State between the meta-*t*Bu of phenyl group on the NHC and the *t*Bu on the Py will be smaller, if the rotate the meta-*t*Bu-phenyl group about 180 degrees

(or remove the tBu to the other meta position). Please compare the ground state energy levels of the two molecular geometries, and also their influence on the $\angle\text{C-N-C-C}$, $\angle\text{C-Pt-N-N}$ and the Transition State.

Response: We greatly appreciate this comment. As you commented, we have done the DFT simulation of PtON-tb-MTB is conducted as presented in the theoretical simulation part. Despite the ground state energy levels between PtON-tb-MTB and PtON-tb-DTB cannot be compared due to significant energy difference at the gas phase. However, the calculated gas-phase energy of PtON-tb-MTB and PtON-tb-DTB is -2113.666794 and -2270.980840 Hartrees, respectively. These large energy variations originated from the *meta*-phenyl substitution of the *t*-butyl group. Moreover, it is enough to understand the steric hindrance effect caused by the *t*-butyl group through the dihedral angle, where the calculated $\angle\text{C1-N1-C2-C3}$ and $\angle\text{C4-Pt-N2-C5}$ are 42.5° and 23.3° , respectively. The decreased $\angle\text{C1-N1-C2-C3}$ of the benzimidazolium carbene linked *meta*-*t*-butyl phenyl part is effectively by reduced steric hindrance due to the removal of the *t*-butyl group and the increased $\angle\text{C4-Pt-N2-C5}$ of the rigid pyridine-carbazole ligand parts. The PtON-tb-DTB has a freely rotatable phenyl ring which affects the excited state geometry due to the reduced steric hindrance by removing the *t*-butyl group. In addition, the dihedral angle ($\angle\text{C1-N1-C2-C3}$) difference between PtON-tb-DTB and PtON-tb-MTB, is indicated by the purple color. The $\angle\text{C6-N3-C5-N2}$ at the transition states of PtON-tb-DTB and PtON-tb-MTB are -90.7° and -109.6° , respectively, which means that the *t*-butyl strongly contributes to tuning the distorted geometry in the transition state. These simulation results are presented with supporting information and additional explanation is presented on the main manuscript.

Furthermore, a QC simulation was performed on PtON-tb-MTB to elucidate the changes in T_1 geometry and transition state following the replacement of the *t*-butyl group with a phenyl ring on the benzimidazolium carbene ligand. The calculated $\angle\text{C1-N1-C2-C3}$ and $\angle\text{C4-Pt-N2-C5}$ values of PtON-tb-MTB at T_1 states are 42.5° and 23.5° , respectively. It means that the free rotation motion due to the removal of the *t*-butyl group can affect the geometrical change in the T_1 state. The angle between PtON-tb-MTB and $\angle\text{C6-N3-C5-N2}$ in the transition state is -109.6° , which is lower than the value of -90.7° for PtON-tb-DTB. This shows that the geometry of the transition state can be adjusted by adding the *t*-butyl group to the benzimidazolium carbene substituted phenyl ring. This is because of the steric hindrance between the *t*-butyl group on the phenyl ring and the pyridine moiety. As a result, the PtON-tb-MTB may not be a good and desired molecule, although it has a higher $\angle\text{C4-Pt-N2-C5}$ due to significantly reduced steric-hindrance, which can cause severe vibrational relaxation and the simulation results are presented in Figure S28.^[16]

	$\angle\text{C1-N1-C2-C3}$ (T_1)	$\angle\text{C4-Pt-N2-C5}$ (T_1)	$\angle\text{C6-N3-C5-N2}$ (TS)
PtON-tb-MTB	42.5°	23.3°	109.6°
PtON-tb-DTB	54.7°	20.3°	90.7°

Figure S28. DFT simulation results of PtON-tb-MTB and PtON-tb-DTB.

6. Lines 204–205, “As previously shown, adding the *t*-butyl group at the meta-position of the ether linkage phenyl ring increases conjugation.” In my opinion, the *t*-butyl is an electron-donating group, which can increase the electron density of the HOMO in PtON-tb-DTB compared to PtON-TBBI, this enable the PtON-tb-DTB to be easier to oxidate, thus, PtON-tb-DTB (–5.50 eV) has shallower HOMO level than that of PtON-TBBI (–5.55 eV). Adding the *t*-butyl group at the meta-position of the ether linkage phenyl ring hardly change the molecular geometry of the Ph-Carbene moiety, please carefully consider whether the “increases conjugation” is reasonable?

Response: Thanks for your valuable suggestion. As the reviewer commented, the shallower HOMO of PtON-tb-DTB can be attributed to donating properties of the *t*-butyl group. Our QC simulation did not show the difference in HOMO based on the *t*-butyl group attached. In addition, PtON-tb-TTB has shown not only a shallower HOMO but also a deeper LUMO than PtON-TBBI in agreement with the QC simulation and experiment. These changes originate from the hyperconjugation effect. Also, the detailed hyperconjugation effect in the optoelectronic material has already been reported [*J. Mater. Chem. C*, 2023, 11, 7030-7038]. These reports illustrate the effectiveness of the hyperconjugation effects of “Si” atoms. However, due to the smaller size of the “C” atom, hyperconjugation effects can be even more effective.

This statement is clarified in the manuscripts at the QC simulation part and experimental photo-physical measurements as follows:

*As illustrated in Figure 1 the highest occupied molecular orbital (HOMO), lowest unoccupied molecular orbital (LUMO) and transition state between T_1 and 3MC state. The Calculated HOMO/ LUMO energy levels of PtON-TBBI, PtON-tb-DTB, and PtON-tb-TTB are -5.57/ -2.40 eV, -5.52/ -2.44 eV, -5.54/ -2.43 eV, respectively. It is worth noting that PtON-tb-TTB with a *t*-butyl substituent on the meta-position of the ether linkage phenyl ring attributed shallower HOMO and deeper LUMO than PtON-TBBI because of the hyperconjugation effect.^[44]*

The measured HOMO/LUMO values of PtON-TBBI, PtON-tb-DTB, and PtON-tb-TTB are -5.55/-2.70 eV, -5.50/-2.71 eV, and -5.53/-2.71 eV, respectively. Both the PL spectra and HOMO/LUMO data show a similar trend as expected by the QC simulation.

*Upon adding the *t*-butyl group at the meta-position of the ether linkage phenyl ring, LUMO of PtON-tb-TTB is lower than PtON-TBBI due to hyperconjugation.*

7. As previous reports (*J. Appl. Phys.* 2000, 87, 8049; *J. Am. Chem. Soc.* 2017, 139, 9783 et, al.), the OLEDs containing TTA process typically exhibit nonlinear Luminance vs current density (L–J) characteristics, therefore, please provide the L–J curves of all the three devices in Figure 5. This might be a direct experiment evidence for the existence of TTA at the current densities of the device lifetime measurements.

Response: As the reviewer commented, device roll-off characteristics are related to L–J curves as reported in *J. Appl. Phys.* 2000, 87, 8049, However, it is hard to say that only TTA can be seen through the L–J curve because TTA and TPA occur through the energy transfer process between excess carriers and excitons. While the current density has increased, both TTA and TPA have increased. Therefore, the L–J curve has almost a similar meaning to the EQE–J curve. In addition, it causes duplicated contents in the paper. Hence, the L–J curve is provided in the supporting information (Figure S27), and we have revised the manuscript as follows:

Figure S27. Luminance (L) vs Current Density (J) graph and Luminescence decline rate (at J=100mA/cm²) of PtON-TBBI, PtON-tb-DTB, and PtON-tb-TTB

“Also, non-linear luminescence (L)-J curve, which clarify roll-off characteristics is presented on the **Figure S27**.^[45] Luminescence decline rate of PtON-TBBI, PtON-tb-DTB, and PtON-tb-TTB was calculated at J= 100 mA/cm² through $\frac{L(\text{ideal})-L(\text{exp})}{L(\text{ideal})} \times 100(\%)$.

The calculated values of PtON-TBBI, PtON-tb-DTB, and PtON-tb-TTB are 44.4%, 38.7%, and 44.2%, respectively. A similar tendency was observed in the EQE-J graph.”

- Please add the excited lifetime values of the films in Figure 6, which can help the readers to check the radiative rates and non-radiative rates of the films. By the way, the non-radiative rate of PtON-tb-TTB (Line 357) should be incorrect.

Response: We appreciate these comments with our mistake. The revised exciton lifetimes of the film states data provided in Figure 6(a) and we have revised the manuscript as follows.

The TRPL measurement was used to measure the exciton lifetime. The exciton lifetime of PtON-TBBI, PtON-tb-DTB, and PtON-tb-TTB were measured to be 1.98, 1.98, and 2.04 μs, as shown in **Figure 6 (a)**, respectively. By using the exciton lifetime and PLQY values, k_r^T and k_{nr}^T were calculated through equation (1). The k_r^T/k_{nr}^T values for PtON-TBBI, PtON-tb-DTB, and PtON-tb-TTB were $4.24 \times 10^5 / 8.08 \times 10^4$, $3.69 \times 10^5 / 1.36 \times 10^5$, and $4.41 \times 10^5 / 4.90 \times 10^4$, respectively.

- For the PtON-TBBI-based blue OLEDs, the device lifetime LT95 was 20.2 h at L0 of 1200 nit, which was estimated about 28.0 h. This device lifetime was much shorter than that of the previous report with LT 95 of about 150 h (Nat. Photonics 2022, 16, 212.) using the same device structure with different thickness of some functional layers. The possible reasons should be discussed in the manuscript. How about the device lifetime using the same thickness of functional layers as the reference (ITO (50 nm)/ HATCN (10 nm)/ PCBBiF (60 nm)/ SiCzCz (5 nm)/ 60 wt% SiCzCz: 27 wt% SiTrzCz: 13 wt% Pt(II)dopant (35 nm)/ mSiTrz (5 nm)/ mSiTrz: Liq (5:5) (35 nm)/ LiF (1.5 nm)/ Al (100 nm))?

Response: Thanks for your grateful comments. We already tried accordingly to reproduce the lifetime of the PtON-TBBI device as followed by the report by using the same common layer materials and similar layer thickness. The device configuration of our optimized structure is ITO (50 nm)/ HATCN (7 nm)/ PCBBiF (45 nm)/ SiCzCz (10 nm)/ 53 wt% SiCzCz: 35 wt% SiTrzCz: 12 wt% Pt(II)dopant (40 nm)/ mSiTrz (5 nm)/ mSiTrz: Liq (2:8) (35 nm)/ LiF (1.5 nm)/ Al (100 nm). Driving voltage, CIE coordinate, and efficiency are very similar to their device. However, it could not reproduce a similar device's lifetime. This reproducible

lifetime issue might originate from the experimental environment and evaporation equipment. However, devices of PtON-TBBI, PtON-tb-DTB, and PtON-tb-TTB are evaluated on the same condition. As a result, the PtON-tb-DTB device shows relatively more stable characteristics than PtON-TBBI and PtON-tb-TTB. Thus, the manuscript is revised as follows:

Despite having nearly identical driving voltage, efficiency, and color coordinates to those earlier reports, the PtON-TBBI device has a substantially different device lifetime.^[33] This divergence may originate from different experimental environments and evaporation equipment. In contrast, the PtON-tb-DTB device displayed 169.3 h while being estimated to be 235.1 h at 1000 nit under the same condition.

10. In Table 3, the acceleration factor ($n = 1.8$) might be overestimated for Pt(II) complexes-based blue OLEDs, it is suggested that the author obtain actual n values through experiments according previous reports (Adv. Mater. 2017, 29, 16.5002; Nat. Photonics 2021, 15, 230.). The experiments are not complicated.

Response: We appreciate and apologise for the reviewer's comments. Even though the acceleration factor of the 1.8 used in the Pt(II) complex can be considered an overestimated value by comparing the reports [Nat. Photonics 2021, 15, 230], we introduced this number because of generally used in the blue devices (ref. 38, 39, 46). We apologize and regretfully say that to conduct further experiments, we will need to synthesize the common layer and dopant materials again, which will take some time to measure the lifetime based on luminescence. Further details regarding the above were revised in the manuscript as follows:

To compare the device lifetime of the PtON-TBBI device with earlier reports, an acceleration factor of 1.8 was used to consider widely reported values.^[38, 39, 46] Although 1.8 of the acceleration factors is utilized, a lifetime of PtON-TTBI at 1000 nit is estimated to take about 28.0 h.

11. In Supporting Information, the chemical structures of ligands 2, 4 and 5 are incorrect, please revise them.

Response: As reviewer suggested, we revised the chemical structures (ions) of ligand 2, 4 and 5 in **Scheme S1**.

12. For all HRMS data in Supporting Information, the $[M+H]^+$ data should be given to correspond to experimental values (found values).

Response: We revised the $[M+H]^+$ or $[M]^+$ data for corresponding material HRMS data in supporting information as follows.

- For synthesis of PtON-TBBI (Pt-ref): **HRMS (QToF, m/z): $[M+H]^+ = 890.3398$ calcd for $C_{48}H_{46}N_4O$, found 890.3396.**
-
- For Synthesis of 2-(3-(1*H*-benzo[*d*]imidazol-1-yl)-5-(*tert*-butyl)phenoxy)-9-(4-(*tert*-butyl)pyridin-2-yl)-9*H*-carbazole, Intermediate material (3) : **HRMS (QToF, m/z): $[M+H]^+ = 565.2967$ calcd for $C_{38}H_{36}N_4O$, found 565.2971.**
-
- For synthesis of 1*H*-Benzimidazolium, 3-[3-(1,1-dimethylethyl)-5-[[9-[4-(1,1-dimethylethyl)-2-pyridinyl]-9*H*-carbazol-2-yl]oxy]phenyl]-1-[3-(1,1-dimethylethyl)phenyl]-, 1,1,1-trifluoromethanesulfonate, Intermediate material (4) : **HRMS (QToF, m/z): 697.3901 calcd for $[C_{48}H_{49}N_4O]^+$, found 697.3903.**
-
- For synthesis of PtON-tb-DTB (Pt-1): **HRMS (QToF, m/z): $[M+H]^+ = 890.3398$ calcd for $C_{52}H_{54}N_4O$, found 890.3392.**
- For synthesis of 1*H*-Benzimidazolium, 1-[3,5-bis(1,1-dimethylethyl)phenyl]-3-[3-[[9-[4-(1,1-dimethylethyl)-2-pyridinyl]-9*H*-carbazol-2-yl]oxy]-5-(1,1-dimethylethyl)phenyl], 1,1,1-

trifluoromethanesulfonate, Intermediate material (5) : HRMS (QToF, m/z): 753.4527 calcd for $[C_{52}H_{57}N_4O]^+$, found 753.4542.

- - For synthesis of PtON-tb-TTB (Pt-2): HRMS (QToF, m/z): $[M+H]^+ = 946.4018$ calcd for $C_{52}H_{54}N_4OPt$, found 946.4024.
 -
13. Considering the high requirements of this journal, please provide the characterization data of ligands 4 and 5.

Response: We have added the characterization data as follows. (1H -NMR, ^{13}C -NMR, and HRMS) of ligands 4 and 5 in the supporting information.

Synthesis of 1H-Benzimidazolium, 3-[3-(1,1-dimethylethyl)-5-[[9-[4-(1,1-dimethylethyl)-2-pyridinyl]-9H-carbazol-2-yl]oxy]phenyl]-1-[3-(1,1-dimethylethyl)phenyl]-, 1,1,1-trifluoromethanesulfonate, Intermediate material (4):

To a three-neck round bottom flask were added compound (3) (7.5 g, 13.28 mmol), (3-tert-butylphenyl)(mesityl)iodonium trifluoromethane sulfonate (10.48 g, 19.92 mmol), copper(II) acetate (0.23 g, 1.32 mmol), and DMF (65 mL). The reaction mixture was stirred at 130°C for 12 h. After cooling to room temperature, the reaction mixture was diluted with water and extracted with ethyl acetate. The organic layer was dried over anhydrous $MgSO_4$. After the solvent was evaporated, the crude product was purified by column chromatography on silica gel (methylene chloride: acetone = 1:10) to afford the desired product as pale yellow (6.4 g, 53 %). 1H -NMR (300 MHz, $CDCl_3$): δ (ppm) = 10.08 (s, 1H), 8.61 (d, $J = 6.0$ Hz, 1H), 8.15 (t, $J = 9.0$ Hz, 2H), 7.79 – 7.62 (m, 12H), 7.48 – 7.28 (m, 5H), 7.16 (d, $J = 9.0$ Hz, 1H), 7.05 (s, 1H), 1.41 (m, 27H). ^{13}C -NMR (300 MHz, $CDCl_3$): δ (ppm) = 163.4, 162.5, 159.5, 157.1, 154.4, 151.4, 149.5, 140.9, 140.8, 140.1, 133.0, 132.2, 131.8, 131.7, 130.4, 128.3, 128.2, 128.0, 125.9, 123.8, 122.6, 122.2, 121.4, 121.1, 120.1, 119.1, 118.3, 117.9, 117.1, 116.3, 114.0, 113.8, 113.4, 110.6, 110.5, 103.3, 35.5, 35.2, 35.1, 31.1, 31.0, 30.5. HRMS (QToF, m/z): 697.3901 calcd for $[C_{48}H_{49}N_4O]^+$, found 697.3903.

Synthesis of 1H-Benzimidazolium, 1-[3,5-bis(1,1-dimethylethyl)phenyl]-3-[3-[[9-[4-(1,1-dimethylethyl)-2-pyridinyl]-9H-carbazol-2-yl]oxy]-5-(1,1-dimethylethyl)phenyl]-, 1,1,1-trifluoromethanesulfonate, Intermediate material (5):

The synthetic procedure for Intermediate material (5) was identical to that of Intermediate material (4), except that (3,5-di-tert-butylphenyl)(mesityl)iodonium trifluoromethane sulfonate was used instead of (3-tert-butylphenyl)(mesityl)iodonium trifluoromethane sulfonate. (Yield: 51 %) 1H -NMR (300 MHz, $CDCl_3$): δ (ppm) = 10.03 (s, 1H), 8.62 (d, $J = 6.0$ Hz, 1H), 8.15 (t, $J = 9.0$ Hz, 2H), 7.78 – 7.53 (m, 12H), 7.48 – 7.42 (m, 6H), 7.37 – 7.28 (m, 4H), 7.17 (d, $J = 9.0$ Hz, 1H), 7.03 (s, 1H), 1.43-1.37 (m, 36H). ^{13}C -NMR (300 MHz, $CDCl_3$): δ (ppm) = 163.3, 159.5, 157.0, 154.4, 154.0, 151.4, 149.5, 140.9, 140.8, 140.1, 133.0, 132.0, 131.7, 128.0, 127.9, 125.9, 125.2, 123.8, 121.4, 121.1, 121.0, 120.0, 119.6, 119.1, 117.9, 117.3, 116.3, 113.9, 113.5, 110.7, 110.6, 103.3, 35.6, 35.3, 35.1, 31.2, 31.0, 30.5. HRMS (QToF, m/z): 753.4527 calcd for $[C_{52}H_{57}N_4O]^+$, found 753.4542.

Other minor issues:

14. "N-heterocyclic carbene", "t-butyl", "N-([1,1'-biphenyl]-4-yl)-9,9-dimethyl-N-(4-(9-phenyl-9H-carbazol-3-

yl)phenyl)-9H-fluoren-2-amine", "9-(3-(triphenylsilyl)phenyl)-9H-3,9'-bicarbazole", "9,9'-(6-(3-(triphenylsilyl)phenyl)-1,3,5-triazine-2,4-diyl)bis(9H-carbazole)", ... the letters "N", "t" and "H" should professionally be italics. Please double check the whole manuscript and Supporting Information and revise them.

Response: As the reviewer suggested, the letters are changed to italics and the changes revised the supporting information as follows.

"Where dipyrzino[2,3-f:2',3'-h]quinoxaline-2,3,6,7,10,11-hexa carbonitrile (HATCN), *N*-([1,1'-biphenyl]-4-yl)-9,9-dimethyl-*N*-(4-(9-phenyl-9*H*-carbazol-3-yl)phenyl)-9*H*-fluoren-2-amine (PCBBiF) were used as hole injection and transporting layer,^[37] respectively 9-(3-(triphenylsilyl)phenyl)-9*H*-3,9'-bicarbazole (SiCzCz) was used as exciton blocking layer and hole transporting p-type host in emissive layer. 9,9'-(6-(3-(triphenylsilyl)phenyl)-1,3,5-triazine-2,4-diyl)bis(9*H*-carbazole) (SiTrzCz) was used for electron transporting n-type host. 2-phenyl-4,6-bis(3-(triphenylsilyl)phenyl)-1,3,5-triazine (mSiTrz) was used as hole blocking layer and electron transporting layer in combination with 8-quinolinolato lithium (Liq)."

15. Please revise "LT 5 0", "LT 9 5" ... as "LT₅₀", "LT₉₅"... in the whole manuscript.

Response: As reviewer suggested, the manuscript is revised accordingly.

16. Please revise "3MLCT", "3LC", "3MC" ... as "³MLCT", "³LC", "³MC" ... in the whole manuscript, also in Figure 6.

Response: As reviewer suggested, the manuscript and Figure 6 were revised. Please see revised Figure 6.

17. Lines 56–58, please revise "Ir" as "Ir(III)", which is consistence with "Pt(II)".

Response: As reviewer suggested, it was revised in the manuscript.

18. Lines 67–68, please add the reference; line 68, revise "and" as "containing".

Response: This part is revised through your comment 3.

As the reviewer suggested, it was revised in the manuscript.

19. Line 59, "low color purity caused broad spectrum" should be "low color purity caused by broad spectrum"?

Response: As reviewer suggested, it was revised in the manuscript.

20. Line 60, "small" is better than "narrow" for FWHM? Typically, "narrow" is used with "spectrum".

Response: As reviewer suggested, Generally, narrow FWHM and narrow spectrums are well used. So, a narrow FWHM is suitable in this manuscript.

21. Line 87, "EQEs" should be "EQE"?

Response: It is revised in the manuscript.

22. Line 91, "tetradentate-based Pt(II) complexes" should be "tetradentate NHC-based Pt(II) complex" or "tetradentate NHC ligand-based Pt(II) complex"?

Response: It is revised in the manuscript.

23. Lines 94–97, "In this study, we introduce two new Pt(II) complexes, PtON-tb-DTB and PtON-tb-TTB, ... These complexes exhibited high PLQYs of 78%, 99%, and 95%, respectively." The statement confuses me, the two complexes are incompatible with three PLQY values, please revise it.

Response: It is revised in the manuscript.

“These Pt(II) complexes of PtON-tb-DTB and PtON-tb-TTB exhibited high PLQYs of 78% and 99% in the 5wt% doped PMMA film, respectively.”

24. It will be clearer if the black background is changed to white in Figure 1.

Response: It is revised in the manuscript **Figure 1**.

25. Line 142, “The decreased \angle C-N-C-C of PtON-tb-DTB influenced, increased conjugation length.” is difficult to understand.

Response: We appreciate the reviewer’s comments. We changed to enhanced conjugation. We think this is better as you suggested. We revised our manuscript as follows.

PtON-tb-DTB's reduced \angle C1-N1-C2-C3 enhanced conjugation in contrast to the other complexes such as PtON-TBBI and PtON-tb-TTB, respectively. It was clarified through the bond length of N1-C2.[47] The N1-C2 values of PtON-TBBI, PtON-tb-DTB, and PtON-tb-TTB are 1.421, 1.417, and 1.420 Å, respectively.

26. Typically, the “Measurements” section should be placed before the “synthesis” section in Supporting Information.

Response: We have done so accordingly in the supporting information part.

27. Line 214, “Clearly, the new materials exhibit a longer lifetime than PtON-TBBI.” should be “Clearly, the new materials exhibit longer lifetimes than that of PtON-TBBI.”?

Response: We revised it accordingly in the manuscript.

28. Lines 272 and 274, The “Figures 4(a) and S17” should be “Figures 4(a) and S16”.

Response: It was revised in the manuscript.

29. Line 326, “N-type host” should be “n-type host”.

Response: It was revised in the manuscript.

30. Line 337, “However, the EQEs of PtON-tb-DTB devices were comparatively lower than that of PtON-TBBI.” should be “However, the EQE of PtON-tb-DTB device was comparatively lower than that of PtON-TBBI.”?

Response: It is revised in the manuscript.

31. The unit of time in Figure 2c should be “ μ s”, not “us”, the same questions are also in Figure 4a and Figure 6a.

Response: We appreciate your considerate comments. It is revised in the manuscript.

Reviewer #2 (Remarks to the Author):

This article studied the effect of the substitution position of the t-butyl group in reference Pt(II) complexes on EL

performances of the fabricated phosphorescent blue OLEDs based on PtON-tb-TTB and PtON-tb-DTB. They gave detailed analysis and it can be found that the substitution position indeed has large effect on the efficiency and lifetime of the fabricated devices. They attributed the large improvement to the decreased TTA process and better hot exciton stability. This work is still very meaningful for further designing and synthesizing high-efficiency and long lifetime blue phosphorescent OLEDs materials. However, as we see, the basic structure of the types of materials has been reported [Nat. Photonics, 16, 212-218 (2022)], where excellent EL efficiency and lifetime have been obtained, although the authors provided a more detailed analysis of material design and influencing factors in this article. Therefore, the article lacks sufficient novelty to be considered for publication in NC. I suggest to submit this article to a more professional magazine. At the same time, the authors should also consider the following issues:

Response: We appreciate the reviewer's comments and understand the reviewer's concerns. As you know, PtON-TBBI blue device with high stability was reported on [Nat. Photonics, 16, 212-218 (2022)]. The characteristics of the requirements for stable blue phosphorescent dopant are highly desirable science. In the previous Nat. Photonics report, the roll of host stability was emphasized for the long device lifetime, but the authors didn't suggest the key factors for designing a Pt (II) complex with high stability. In this manuscript, we presented a new Pt(II) complex material design and achieved a longer device lifetime of stable blue phosphorescent OLED, along with the depth analysis of device degradation and the experimental results. Moreover, the stable blue Pt(II) complex design has not been much investigated until now. Thus, the concept proof of this manuscript could be more useful in establishing new materials for display applications in the future.

1. "lifespan" is usually written as "lifetime".

Response: We appreciate the reviewer's suggestion. As the reviewer commented, "lifespan" is changed to "lifetime" in the revised manuscripts.

2. In abstract, the 169.3 h lifetime should be TL95 at 1200 cd/m², which should be clearly stated.

Response: We appreciate the reviewer's suggestion lifetime of the PtON-tb-DTB device be mentioned in the abstract more clearly.

"Additionally, the PtON-tb-DTB device showed exceptional operational stability, with a lifetime of 169.3 h at initial luminance of 1200 nit".

3. As shown, PtON-tb-DTB emitted the lowest PLQY and EL efficiency, but have the longest lifetime than PtON-TBBI and PtON-tb-TTB, which is usually not easy to understand.

Response: We appreciate the reviewer's comments and understand the reviewer's concerns. As the reviewer commented, a device's lifetime can be proportional to its efficiency due to its required current value. However, it is not always proportional, and it depends on the various factors that are related to the device's lifetime. Accordingly, similar device performance was reported with blue phosphorescent and TADF OLED lifetimes. In the "J. Mater. Chem. C, 2021, 9, 17412–17418", the HT3: ET8 host employed device shows 21.9% of EQEmax and 30 h of LT95. However, the HT4: ET8 used device exhibited a higher EQEmax of 22.4 with a shorter LT95 of 17 h, and they used a similar IrE dopant. Similarly [Adv. Optical Mater, 2022, 10, 2102309], the CzTrzBp-used device shows 30.5 h of LT80 and an EQEmax of 9.2%. However, the CzTrzPh-used device shows 17.2 h of LT80 and an EQEmax of 12.5%.

4. The energy level diagrams of PtON-tb-DTB, PtON-TBBI and PtON-tb-TTB are necessary to be given by calculating or measuring.

Response: Thank you for your comment. Previously, we presented an approximate energy diagram of dopants on Figure S24 due to almost similar HOMO and LUMO energy level. As you suggested, the measured HOMO and LUMO energy level of dopants is presented in Figure S24, respectively.

5. What is the physical basis of Figure 6(c). As we see, the energy level of S0 in PtON-TBBI is completely different from that of PtON-tb-DTB.

Response: We appreciate the reviewer's insightful comments. It was our simple mistake. The corrected Figure 6(c) we revised the manuscript as follows.

Figure 6. (a) TRPL measurements for 12 wt% doped films on SiCzCz and SiTrzCz mixed host. (b) Device lifetime of PtON-TBBI, PtON-tb-DTB, and PtON-tb-TTB at 1200 nit and (c) Schematic diagram of degradation mechanism based on RMSD simulation.

Reviewer #3 (Remarks to the Author):

This manuscript reports the approach of introducing tert-butyl group in an appropriate position to suppress MMLCT in platinum(II)-based emitters to achieve high PLQY of up to 99% in doped films and for improving OLED EQEs up to 26.3%. Extensive computational studies, involving MD, QM, were used to calculate intermolecular distance, rate of excited state processes and deactivation processes, and behaviour in the solid state. Even though it is known that increasing the steric bulk of the square planar emitters can suppress intermolecular interaction, this work has demonstrated that the placement of these tert-butyl group must be strategic in controlling the various excited state processes for performance enhancement. I would recommend this work to be published in nature communication provided the following issues are addressed.

Response: We appreciate the referee for recommending publishing this manuscript in Nature Communication after revision. The issues indicated by reviewer have been addressed, as described in the following responses.

1. MS lines 130-131, how does the removal of one tert-butyl group on the benzimidazole carbene and the flexibility created by such modification correlate with the formation of MMLCT?

Response: We thank the reviewer's suggestion. The removal of the *t*-butyl group on the phenyl ring reduces the steric hindrance, which causes a decreased $\angle C1-N1-C2-C3$. On the other hand, the reduced dihedral angle of $\angle C1-N1-C2-C3$ can cause MMLCT due to increased planarity. However, simultaneously increased $\angle C4-Pt-N2-C5$ in excited state seems to suppress the MMLCT formation in our research by reducing the interaction with d_{z^2} of Pt. In other words, the PtON-tb-DTB, one *t*-butyl group is removed from the benzimidazolium carbene substituted phenyl ring, which reduces steric hindrance and makes the moiety more flexible and rotatable, which can effectively hinder the formation of MMLCT and reduce the $E_{a,3MC \rightarrow T1}$. The related data was provided in **Table 1**.

To be clear in our description, we have added this part in our manuscript as follows:

"This indicates that because of the lower steric hindrance, PtON-tb-DTB has a more distorted conformation than PtON-TBBI through the freely rotating motion of bulky substitution. The PtON-tb-DTB possesses a larger dihedral angle at the T_1 state, which is expected to alleviate the formation of the MMLCT by reducing the intermolecular interaction between the vacant d_{z^2} orbitals of Pt(II) (central metal atom). Thus, the formation of MMLCT could be more effectively suppressed by PtON-tb-DTB than that of the PtON-TBBI dopant."

2. MS Lines 137-138, Can you elaborate more on how the conjugation length is changed by the introduction of the tert-butyl group on the meta position of tether linkage phenyl ring? I don't see much difference in the calculated structure.

Response: Thanks for the reviewer's comment. Firstly, PtON-tb-TTB has a shallower HOMO level of -5.53 eV than PtON-TBBI. In addition, the LUMO energy level of PtON-tb-TTB is deeper than that of PtON-TBBI, although the difference is small. Further, there is a similar tendency observed in the DFT calculation for these materials. As a result, the decreased band gap of PtON-tb-TTB is affected by hyperconjugation effects that are induced by the *t*-butyl group at the *meta*-position of the ether linkage phenyl ring. In [J. Mater. Chem. C, 2023, 11, 7030–7038], the authors reported the hyperconjugation effects of "Si" atoms in EBL material. Generally, the "C" atom shows a higher contribution to hyperconjugation.

To be clear in our description, we have changed our manuscript as follows:

*PtON-tb-TTB with *t*-butyl substituent on the meta-position of ether linkage phenyl ring attributed shallower HOMO and deeper LUMO than PtON-TBBI due to hyperconjugation effects.^[44]*

3. MS Fig 2b, since the spectra of two complexes are shifted relative to the reference. Would it be better to unlabel the x-axis?

Response: We appreciate your considerate comments. As you recommended, x-axis is unlabeled in **Figure 2b**. we have done accordingly in the manuscript.

Figure 2. (a) UV-visible spectroscopy in methylene chloride (1.0×10^{-5} M) and photoluminescence spectroscopy in 5wt% doped PMMA of PtON-TBBI, PtON-tb-DTB, and PtON-tb-TTB (b) Shifted spectra of PtON-tb-DTB and PtON-tb-TTB to that of PtON-TBBI for comparison of the vibrational bands. (c) Measurements of time-resolved photoluminescence (TRPL) spectra by using 5wt% doped on the PMMA matrix.

4. MS Fig3 and Lines 247-248, is the peak intensity difference between 5 wt% and 50 wt% doped films for PtON-tb-DTB very significantly different from that of PtON-tb-TTB?

Response: Thank you for your comments. In these lines 247-248, “peak intensity difference” means second vibrational peak difference presented as 0.054 and 0.065 of PtON-tb-DTB and PtON-tb-TTB, respectively. It does not show much difference between PtON-tb-DTB and PtON-tb-TTB. However, it can be confusing that both materials and it shows significantly difference in the second vibrational peak intensity difference. Therefore, we revised the main manuscript as follows

*“Further, investigating the formation of MMLCT, the PL spectra of PMMA films doped with 5 wt% and 50 wt% of the materials were measured as shown in **Figure 3**. The differences in the second vibronic peak intensity between the 5 wt% and 50 wt% doped films of PtON-TBBI, PtON-tb-DTB, and PtON-tb-TTB are 0.211, 0.054, and 0.065, respectively. As we expected from QC and MD simulation, the MMLCT formation of PtON-tb-TTB was significantly suppressed owing to the additional t-butyl group. In addition, PtON-tb-DTB shows suppressed MMLCT formation compared with PtON-TBBI, although it has a similar density value in MD simulation. Our results imply that orbital overlap reduction plays a role in MMLCT formation as well. Thus, PtON-tb-DTB has a lower second vibrational peak intensity difference between the 5 wt% and 50 wt% doped films than that of PtON-tb-TTB, and this difference is induced by the higher dihedral angle ($\angle C4-Pt-N2-C5$) of PtON-tb-DTB. This experiment shows that the dihedral angles ($\angle C4-Pt-N2-C5$) can also be crucial parameters to suppress MMLCT formation, besides intermolecular distance.”*

5. MS Lines 272-273, the absorption of these complexes at 340 nm is not really that poor based on the UV spectra. But how exactly can one avoid bimolecular exciton quenching by choosing this wavelength?

Response: We appreciate you for your thoughtful remarks. The wavelength of 340 nm has relatively low absorption in the wavelength range below 380 nm. The normalized absorption values of all the material is 0.27-0.29, and it is decreased below 380 nm than 0.27-0.29. However, we were concerned that a longer excitation wavelength than 380 nm can cause the spectrum to mix between excitation and emission wavelength; therefore, the 340 nm excitation wavelength was selected for our research. It is difficult to say that bimolecular quenching is completely avoided, but the relationship between the measured and expected PLQY is well-matched. The expected PLQY is calculated using the obtained FRET and DET rates. Respective FRET and DET rate fit well with their equations as presented in **Figure 4 (b)** and **(c)**. Therefore, we believe that 340 nm of wavelength is sufficiently weak to see exciton diffusion effects rather than bimolecular quenching.

“To avoid bimolecular exciton quenching induced at high exciton concentrations, a 340 nm excitation wavelength is selected, because it has a relatively low absorption peak below 380 nm of wavelength, where the emission peak does not mix with the excitation wavelength.”

*“In **Figure 4** and **S22**, experimental values are well correlated with our expected values. It means that bimolecular quenching is sufficiently suppressed in this experiment and exciton quenching occurs through FRET and DET process.”*

6. Please provide procedure for result fitting in the supporting information.

Response: The fitting procedure of FRET rate, DET rate, and roll-off analysis is presented on the supporting information as new contents.

Thank you,

Best Regards,

Professor. Jang Hyuk Kwon

REVIEWERS' COMMENTS

Reviewer #1 (Remarks to the Author):

The authors have addressed all the comments and revised the manuscript, I recommend it to publish in Nature Communications.

Reviewer #2 (Remarks to the Author):

The article has made significant improvements after revision, and it may be considered for publication without further modifications.

Reviewer #3 (Remarks to the Author):

The authors have addressed the issues to my satisfaction and it is ready to be published.